# Closed-Form Merging of Parameter-Efficient Modules for Federated Continual Learning

**Riccardo Salami**[1]     **Pietro Buzzega**[1]     **Matteo Mosconi**[1]

**Jacopo Bonato**[2]     **Luigi Sabetta**[2]     **Simone Calderara**[1]

[1]AImageLab - University of Modena and Reggio Emilia, Modena, Italy
`name.surname@unimore.it`

[2]Leonardo Labs, Roma, Italy
`name.surname.ext@leonardo.com`

## Abstract

Model merging has emerged as a crucial technique in Deep Learning, enabling the integration of multiple models into a unified system while preserving performance and scalability. In this respect, the compositional properties of low-rank adaptation techniques (e.g., LoRA) have proven beneficial, as simple averaging LoRA modules yields a single model that mostly integrates the capabilities of all individual modules. Building on LoRA, we take a step further by imposing that the merged model matches the responses of all learned modules. Solving this objective in closed form yields an indeterminate system with $A$ and $B$ as unknown variables, indicating the existence of infinitely many closed-form solutions. To address this challenge, we introduce LoRM, an alternating optimization strategy that trains one LoRA matrix at a time. This allows solving for each unknown variable individually, thus finding a unique solution. We apply our proposed methodology to Federated Class-Incremental Learning (FCIL), ensuring alignment of model responses both between clients and across tasks. Our method demonstrates state-of-the-art performance across a range of FCIL scenarios. The code to reproduce our experiments is available at github.com/aimagelab/fed-mammoth.

## 1 Introduction

Humans naturally excel at learning a diverse array of skills independently, effortlessly acquiring knowledge across multiple domains throughout their lives. In contrast, the traditional paradigm for artificial neural networks relies on training a unified model on a single, large dataset. While this approach facilitates the simultaneous incorporation of different skills, it lacks the capacity for specialized or incremental learning, making it less adaptable and responsive to changes in the environment. To overcome this limitation and mimic human flexibility, various paradigms have been developed to enhance neural networks' ability to manage diverse skills effectively. Multi-Task Learning (Caruana, 1997) involves training a model on several tasks simultaneously, promoting the sharing of representations across tasks, while Continual Learning (CL) (McCloskey & Cohen, 1989) focuses on enabling models to learn tasks incrementally without forgetting prior knowledge. Federated Learning (FL) (McMahan et al., 2017), on the other hand, focuses on decentralized training by distributing data and computation across separate clients, each specializing in their local task. While each of these scenarios has its own unique characteristics, they all share the common objective of integrating task-specific modules into a unified framework.

Recently, large pre-trained architectures have facilitated model editing (Ortiz-Jimenez et al., 2024) and specialization (Bowman et al., 2023), particularly for fine-tuning downstream tasks. In practice, deep models often leave their parameters fixed, leveraging Parameter-Efficient Fine-Tuning (PEFT) techniques to adapt to new tasks effectively. Among PEFT methods, Low-Rank Adaptation (LoRA) (Hu et al., 2022) has emerged as a prominent approach. LoRA introduces residual weights in the form of $\Delta W = BA$, where $B$ and $A$ are low-rank matrices. These residuals, commonly

referred to as task vectors (Ilharco et al., 2023), form the foundation of the novel model merging literature, which has introduced various approaches for their integration. For example, Zhang et al. (2023c) explore the combination of task vectors through linear arithmetic operations, while other works focus on identifying optimal coefficients for weighting these modules during aggregation (Yadav et al., 2024; Yang et al., 2024; Wu et al., 2024). In contrast to these methods, we propose a novel solution that merges LoRA modules in a *closed-form*.

To evaluate the feasibility of our approach, we base our investigation on a well-defined empirical framework, situating our work at the intersection of Federated Learning and Continual Learning. These two paradigms are ideal for assessing the merging of task vectors, as they encompass both spatial (across clients) and temporal (over tasks) aggregation. Specifically, in Federated Class-Incremental Learning (FCIL) (Yoon et al., 2021), tasks are introduced incrementally, and data is distributed across multiple clients in a decentralized manner.

We introduce a novel approach, termed Low-rank Regression Mean (LoRM), tailored to the FCIL setting. Our method builds upon RegMean (Jin et al., 2023), a model merging technique derived from an exactly solvable regression problem. Starting from RegMean's formulation, we develop a strategy to merge LoRA modules in closed form. Our derivations result in two key equations — one for merging matrix $\boldsymbol{A}$ and another one for matrix $\boldsymbol{B}$. During training, we propose an *alternating* optimization procedure, where one matrix is learned while the other remains fixed across models. In the Federated Class-Incremental Learning setting, spatial aggregation across clients is performed using this alternating procedure. Conversely, for temporal aggregation, task-specific modules are merged by applying the RegMean formulation directly to the full residual weights $\Delta \boldsymbol{W}$ of all tasks.

In summary, the key contributions of this work are as follows:

- We explore the feasibility of merging LoRA modules using a closed-form solution.
- We introduce LoRM, a novel FCIL approach that leverages insights from our exploration.
- We demonstrate the effectiveness of our method across diverse datasets and varying degrees of data distribution, achieving state-of-the-art results.

## 2 BACKGROUND AND MOTIVATION

### 2.1 PRELIMINARIES

LoRA (Hu et al., 2022) was introduced to reduce the number of trainable parameters when fine-tuning pre-trained models. Formally, let $\boldsymbol{W}_0 \in \mathbb{R}^{d \times k}$ represent the matrix of pre-trained weights of a linear layer, and let $\boldsymbol{x} \in \mathbb{R}^{k \times 1}$ be the input vector for that layer. The output $\boldsymbol{h}$ is given by:

$$\boldsymbol{h} = \boldsymbol{W}_0 \boldsymbol{x} + \Delta \boldsymbol{W} \boldsymbol{x} = \boldsymbol{W}_0 \boldsymbol{x} + \boldsymbol{B} \boldsymbol{A} \boldsymbol{x}, \tag{1}$$

where $\Delta \boldsymbol{W} = \boldsymbol{B} \boldsymbol{A}$ is the residual weight introduced by LoRA, with matrices $\boldsymbol{A}$ and $\boldsymbol{B}$ as the only components being trained. The efficiency of this approach stems from the low rank $r$ of the matrices, where $\boldsymbol{B} \in \mathbb{R}^{d \times r}$ and $\boldsymbol{A} \in \mathbb{R}^{r \times k}$, with $d$ and $k$ representing the number of output and input features of the layer, respectively. Consequently, the number of trainable parameters is $r \cdot (d+k)$, which, since $r \ll d$, constitutes only a fraction of the $d \cdot k$ parameters required for full fine-tuning. Additionally, $\boldsymbol{B}$ is initialized to 0: *i.e.*, the first forward pass is equivalent to the absence of a LoRA residual.

RegMean (Jin et al., 2023) introduces a method for merging a collection of $N$ linear layers $\{\boldsymbol{W}_i\}_{i=1}^{N}$, each corresponding to $N$ distinct models trained on distinct inputs $\{\boldsymbol{X}_i\}_{i=1}^{N}$. The goal is to identify a single linear layer that produces responses that closely match those of the starting layers. Specifically, the objective function is defined as follows[1]:

$$\text{minimize } \Omega = \sum_{i=1}^{N} \|\boldsymbol{W}_M \boldsymbol{X}_i - \boldsymbol{W}_i \boldsymbol{X}_i\|_2^2. \tag{2}$$

By computing the gradient of $\Omega$ with respect to $\boldsymbol{W}_M$ and setting it equal to zero, a closed-form solution is obtained. Notably, the merged layer $\boldsymbol{W}_M$ is computed as follows:

$$\boldsymbol{W}_M = \left( \sum_{i=1}^{N} \boldsymbol{W}_i \boldsymbol{X}_i \boldsymbol{X}_i^{\top} \right) \left( \sum_{i=1}^{N} \boldsymbol{X}_i \boldsymbol{X}_i^{\top} \right)^{-1}. \tag{3}$$

---

[1]We rework the original formulation to ease subsequent derivations.

In this context, $\boldsymbol{X}_i \in \mathbb{R}^{k \times \text{samples}}$ represents the input to the $i$-th layer being merged. Hence, the Gram matrix $\boldsymbol{X}_i \boldsymbol{X}_i^\top$ has dimensions $k \times k$ features.

While RegMean finds a weight matrix that approximates the outputs of all layers considered, it does not take into account starting from pre-trained weights or the use of low-rank modules (*e.g.*, LoRA). Given these considerations, we ask whether this method can be suitably adapted to merge LoRA modules. In other words:

> *Can we devise a strategy to merge LoRA modules in a closed form?*

To further explore this question from an experimental standpoint, we place our investigation in the context of a Federated Class-Incremental Learning scenario.

## 2.2 PROBLEM SETTING

In Federated Class-Incremental Learning, the dataset $\mathcal{D}$ is first divided into $T$ tasks, each consisting of a distinct set of classes. Then, each partition $\mathcal{D}^t$ corresponding to the $t$-th task is further distributed among $N$ clients, resulting in $\mathcal{D}_i^t$ for the $i$-th client. Similar to standard Class-Incremental Learning (Van de Ven et al., 2022), the task-specific partitions $\{\mathcal{D}^t\}_{t=1,\dots,T}$ arrive sequentially.

In this federated scenario, the training for each task is conducted over multiple communication rounds. During each round, clients are restricted to learning only from their local dataset $\mathcal{D}_i^t$. The local optimization objective for client $i$, based on the loss function $\mathcal{L}$, can be formally expressed as:

$$\underset{\theta_i}{\text{minimize}} \ \mathbb{E}_{(x,y) \sim \mathcal{D}_i^t} \left[ \mathcal{L}(f(x; \theta_i), y) \right], \tag{4}$$

where $x$ and $y$ denote the inputs and corresponding labels, respectively, with $\theta_i$ representing the set of parameters for client $i$, and $f(\cdot; \theta_i)$ denoting the associated model.

After completing local updates, each client sends its model parameters $\theta_i$ to the central server, where they are aggregated with those from other clients. The server then sends the global aggregated model back to the clients, marking the end of a communication round. This process repeats for several rounds until the training for task $t$ is completed. Once all rounds for task $t$ have finished, the system progresses to the next task $t + 1$ using the corresponding dataset $\mathcal{D}^{t+1}$. The ultimate objective is to obtain a global model, derived from the aggregation of local models performed by the server, that functions well across all incremental tasks and successfully integrates the distributed knowledge.

## 3 METHODOLOGY

### 3.1 A CLOSED-FORM SOLUTION FOR LORA MERGING

Using the same notation as in Equation 2, let $\boldsymbol{W}$ denote the weight matrix of a given layer. Since LoRA (see Equation 2) is applied to each linear layer across all clients, we express the weight matrix for the $i$-th client as $\boldsymbol{W}_i = \boldsymbol{W}_0 + \boldsymbol{B}_i \boldsymbol{A}_i$, where $\boldsymbol{W}_0$ is the shared pre-trained weight matrix, and $\boldsymbol{B}_i$, $\boldsymbol{A}_i$ represent the low-rank matrices specific to the client. This formulation is consistent across clients, as they all utilize the same model architecture. At the end of each communication round, after conducting local training on the LoRA matrices, the goal is to merge the corresponding matrices (*i.e.*, $\boldsymbol{A}$'s with $\boldsymbol{A}$'s and $\boldsymbol{B}$'s with $\boldsymbol{B}$'s) using the closed-form solution derived from RegMean. Starting from Equation 2, our objective becomes:

$$\Omega = \sum_{i=1}^{N} \|(\boldsymbol{W}_0 + \boldsymbol{B}_M \boldsymbol{A}_M) \boldsymbol{X}_i - (\boldsymbol{W}_0 + \boldsymbol{B}_i \boldsymbol{A}_i) \boldsymbol{X}_i\|_2^2. \tag{5}$$

To find the optimal $\boldsymbol{A}_M$ and $\boldsymbol{B}_M$ that minimize $\Omega$, we differentiate $\Omega$ with respect to each variable, one at a time, and set the gradients to zero:

$$\begin{cases} \dfrac{\partial \Omega}{\partial \boldsymbol{B}} = 0 \\[2mm] \dfrac{\partial \Omega}{\partial \boldsymbol{A}} = 0 \end{cases} \tag{6}$$

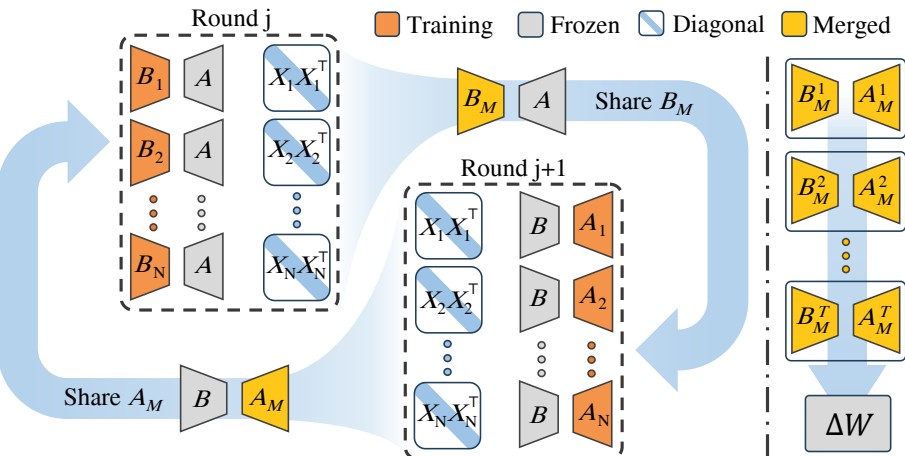

Figure 1: Training and aggregation procedure of LoRM, alternating between rounds. The figure depicts both Federated and Incremental aggregation for a generic layer.

However, the system reveals indeterminate, as the two equations exhibit linear dependence on one another during the calculations. For further mathematical derivations illustrating the infeasibility of this approach, refer to appendix A.1.

As a solution, we propose freezing one of the two matrices. This means that either $\boldsymbol{A}$ or $\boldsymbol{B}$ is shared across clients (*i.e.*, treated as a constant), and a single closed-form equation determines how to merge the trainable matrices across clients. If we choose to share $\boldsymbol{A}$, the merged $\boldsymbol{B}_M$ is obtained as:

$$\boldsymbol{B}_M = \left( \sum_{i=1}^{N} \boldsymbol{B}_i \boldsymbol{A} \boldsymbol{X}_i \boldsymbol{X}_i^\top \right) \boldsymbol{A}^\top \left( \boldsymbol{A} \sum_{i=1}^{N} \boldsymbol{X}_i \boldsymbol{X}_i^\top \boldsymbol{A}^\top \right)^{-1}. \tag{7}$$

Note that $\boldsymbol{A}$ does not have a subscript $i$, as it is identical for all clients. Instead, if we opt to share $\boldsymbol{B}$, the merged $\boldsymbol{A}_M$ is computed as:

$$\boldsymbol{A}_M = \left( \sum_{i=1}^{N} \boldsymbol{A}_i \boldsymbol{X}_i \boldsymbol{X}_i^\top \right) \left( \sum_{i=1}^{N} \boldsymbol{X}_i \boldsymbol{X}_i^\top \right)^{-1}. \tag{8}$$

For the formal derivation of these equations, refer to appendices A.1.1 and A.1.2.

## 3.2 LoRM

Having established closed-form solutions for weight merging, we now outline the full procedure of LoRM, illustrated in Figure 1. Each client $i$ begins by optimizing its own $\boldsymbol{B}_i$, which is learned during local training, and a shared $\boldsymbol{A}$, initialized and distributed by the server. In this first round, it is essential to freeze $\boldsymbol{A}$, as LoRA's $\boldsymbol{B}$ is initialized to 0. Freezing $\boldsymbol{B} = 0$ across all clients would render the training ineffective. At the end of each round, each client $i$ computes the Gram matrix $\boldsymbol{X}_i \boldsymbol{X}_i^\top$ with a forward pass on all examples[2]. Then, it sends $\boldsymbol{B}_i$ and $\boldsymbol{X}_i \boldsymbol{X}_i^\top$ to the server, where the merging operation (as described in Equation 7) is performed. The resulting $\boldsymbol{B}_M$ is then sent back to the clients, who begin the next communication round.

*Alternated optimization.* Empirically, we observe that consistently training only the matrix $\boldsymbol{B}$ while keeping $\boldsymbol{A}$ fixed can be limiting (see Section 4.3 for a detailed analysis). Therefore, to fully exploit LoRA's representational potential, we introduce an *alternating* training approach, where the matrix to be updated changes at each round. Specifically, after the first round, we freeze $\boldsymbol{B}$, which is already synchronized across clients due to the previous server aggregation, and train $\boldsymbol{A}$ instead. Then, at the end of the second round, all local $\boldsymbol{A}$'s are aggregated using Equation 8. This strategy also improves

---

[2]Note that this process provides a Gram matrix for every layer in the network. In this discussion, $\boldsymbol{X}_i \boldsymbol{X}_i^\top$ is related to the single generic linear layer.

---

**Algorithm 1 – LoRM**, for a generic linear layer    Round j    Round j + 1

---

1: **Input:** $T$ tasks; $N$ clients residual modules $\Delta \boldsymbol{W}_i = \boldsymbol{B}_i \boldsymbol{A}_i$; layer's pre-trained weights $\boldsymbol{W}_0$.
2: **for each** task $t \in \{1, \dots, T\}$ **do**
3:    **for each** communication round **do**
4:    **Clients side:**
5:    **for each** client $i \in \{1, \dots, N\}$ **in parallel do**
6:      $\boldsymbol{A}^t = \boldsymbol{A}_M^t$ **or** $\boldsymbol{B}^t = \boldsymbol{B}_M^t$          ▷ Fix either matrix
7:      **for each** epoch **do**
8:        **for each** input $\boldsymbol{x}$ **do**
9:          $\boldsymbol{h} = \boldsymbol{W}_0 \boldsymbol{x} + \boldsymbol{B}_i^t \boldsymbol{A}^t \boldsymbol{x}$ **or** $\boldsymbol{h} = \boldsymbol{W}_0 \boldsymbol{x} + \boldsymbol{B}^t \boldsymbol{A}_i^t \boldsymbol{x}$      ▷ Forward pass
10:          Optimize $\boldsymbol{B}_i^t$ **or** Optimize $\boldsymbol{A}_i^t$
11:        **end for**
12:      **end for**
13:      Send to the server $\mathrm{diag}(\boldsymbol{X}_i^t \boldsymbol{X}_i^{t\top})$
14:      Send $\boldsymbol{B}_i^t$ to the server **or** Send $\boldsymbol{A}_i^t$ to the server
15:    **end for**
16:    **Server side:**
17:      Use Equation 7 and distribute $\boldsymbol{B}_M^t$ **or** Use Equation 8 and distribute $\boldsymbol{A}_M^t$
18:    **end for**
19: **end for**
20: Use Equation 9 to compute the final residual module      ▷ RegMean across tasks

---

efficiency, as only one of the two matrices needs to be communicated per round, offering a significant advantage compared to transmitting the whole LoRA module.

*Merge task-specific modules.* At the conclusion of the generic task $t$, the merged matrices $\boldsymbol{B}_M^t$ and $\boldsymbol{A}_M^t$ are multiplied to obtain the task-specific residual module: $\Delta \boldsymbol{W}^t = \boldsymbol{B}_M^t \boldsymbol{A}_M^t$. On the server side, when aggregating modules from all tasks at the end of the training, we apply the standard RegMean formulation (Equation 3) to merge the aforementioned residuals as:

$$\Delta \boldsymbol{W} = \left( \sum_{t=1}^{T} \boldsymbol{B}_M^t \boldsymbol{A}_M^t \boldsymbol{X}^t \boldsymbol{X}^{t\top} \right) \left( \sum_{t=1}^{T} \boldsymbol{X}^t \boldsymbol{X}^{t\top} \right)^{-1}. \tag{9}$$

To accomplish this, the server stores the task-specific weight $\{\Delta \boldsymbol{W}^1, \dots, \Delta \boldsymbol{W}^T\}$ throughout the training process, along with the global Gram matrices $\{\boldsymbol{X}^t \boldsymbol{X}^{1\top}, \dots, \boldsymbol{X}^t \boldsymbol{X}^{T\top}\}$. Due to the associative property of matrix multiplication, the computation of the latter is straightforward, achieved by summing the client-specific Gram matrices shared by the clients at the end of each task $t$: $\boldsymbol{X}^t \boldsymbol{X}^{t\top} = \sum_{i=1}^{N} \boldsymbol{X}_i^t \boldsymbol{X}_i^{t\top}$. It is worth noting that, for what concerns the classification layer, concatenating the task-specific classification heads is equivalent to applying RegMean. We provide a mathematical derivation of this result in appendix A.4.

The overall procedure yields a final weight residual $\Delta \boldsymbol{W}$ that promotes both generalization across tasks and robustness to knowledge distribution. The pseudocode for LoRM, applied to a generic linear layer, is presented in Algorithm 1.

## 3.3 LoRM Characteristics for Federated Class-Incremental Learning

In Federated Class-Incremental Learning (FCIL), data privacy, efficiency, and the rate of convergence are crucial concerns. LoRM is specifically designed to address these challenges.

*Privacy-preserving.* In FCIL, inputs $\boldsymbol{X}_i$ for the $i$-th client's generic layer cannot be transmitted to the central server, as doing so, especially for the first layer, would compromise data privacy by effectively sharing the dataset. In LoRM, instead of transmitting $\boldsymbol{X}_i$ directly, we send the Gram matrix $\boldsymbol{X}_i \boldsymbol{X}_i^\top$, which obfuscates the original data. Furthermore, only the diagonal of the Gram matrix is communicated, ensuring that no local data is exposed.

*Efficiency.* The use of LoRA inherently improves efficiency compared to full fine-tuning, as it necessitates the communication of only two low-rank matrices for each layer. Additionally, LoRM's alternating optimization procedure permits the transmission of only one matrix per communication round, further reducing overhead. The use of solely the diagonal of the Gram matrix mitigates communication costs even further, as it avoids the need to transmit the full matrix. As a result, at the conclusion of each communication round, each client transmits a low-rank matrix and a vector to the server for each layer.

*Rate of convergence.* Ultimately, LoRM demonstrates faster convergence compared to other FCIL baselines, as further discussed in Section 4.2.

# 4 EXPERIMENTAL STUDY

## 4.1 EVALUATION SETTINGS

*Vision datasets.* The importance of evaluating pre-trained models on datasets that deviate significantly from their pre-training domain is well-established in the literature (Kolesnikov et al., 2020; Kornblith et al., 2019). Accordingly, our experimental section features three *in-domain* datasets, two *specialization* datasets, and one *out-of-domain* dataset. For in-domain evaluation, we use CIFAR-100 (Krizhevsky et al., 2009), ImageNet-R (Hendrycks et al., 2021a), and ImageNet-A (Hendrycks et al., 2021b); to assess specialization within a single category, we utilize *Cars-196* (Krause et al., 2013), and *CUB-200* (Wah et al., 2011). Finally, for out-of-domain evaluation, we employ EuroSAT (Helber et al., 2018), a satellite dataset recognized by Oh et al. (2022) as one of the most challenging for domain adaptation from ImageNet-21k pre-training. CIFAR-100, ImageNet-R, ImageNet-A, CUB-200, and Cars-196 are divided into 10 incremental tasks, while EuroSAT is split into 5 due to its fewer classes. Each task includes an equal share, except for Cars-196, where the final task contains 16 classes. The data is distributed across 10 clients using the commonly adopted *distribution-based* label imbalance setting (Li et al., 2022; Yurochkin et al., 2019), where partitioning is governed by a Dirichlet distribution parameterized by $\beta$. A smaller $\beta$ value corresponds to a more challenging data distribution. We evaluate all methods across three scenarios per dataset, using $\beta \in \{0.5, 0.1, 0.05\}$ for CIFAR-100 and ImageNet-R, and $\beta \in \{1, 0.5, 0.2\}$ for the others to account for fewer examples or classes. For further details on data pre-processing, including the dataset-specific augmentations used, please refer to appendix B.

*Language datasets.* To validate our methodology in a completely different domain, we also include Out-Of-Scope (OOS) (Larson et al., 2019), a textual intent-classification dataset comprising 150 classes. These classes are evenly distributed across 5 incremental tasks, with $\beta \in \{1.0, 0.5, 0.2\}$.

*Evaluated approaches.* We compare LoRM against 10 competing methods spanning different fields. From Continual Learning, we evaluate EWC (Kirkpatrick et al., 2017), LwF (Li & Hoiem, 2017), L2P (Wang et al., 2022b), and CODA-Prompt (Smith et al., 2023). Following previous studies (Zhang et al., 2023b; Guo et al., 2024), we adapt these methods to the federated domain by merging client weights using the FedAvg algorithm (McMahan et al., 2017). We also include FisherAvg (Matena & Raffel, 2022) and RegMean (Jin et al., 2023), from the model merging literature, along with CCVR (Luo et al., 2021) and FedProto Tan et al. (2022) from Federated Learning. These approaches are adapted for Continual Learning by applying Asymmetric Cross-Entropy (ACE) (Caccia et al., 2022), a standard method that optimizes classification heads independently for each task. Additionally, we evaluate two algorithms specifically designed for Federated Class-Incremental Learning: TARGET (Zhang et al., 2023b) and PILoRA (Guo et al., 2024), the latter representing the current State Of The Art. For OOS, we also evaluate TIES-Merging (Yadav et al., 2024) from the model merging literature. Finally, we include Joint results as an upper bound, achieved by training the backbone on the entire dataset without any federated or incremental partitioning.

*Implementation Details.* As the backbone for LoRM and all competing approaches, we employ a ViT-B/16 model (Dosovitskiy et al., 2021) pre-trained on ImageNet-21K (Ridnik et al., 2021) on all vision datasets. We set the number of epochs per communication round to 5 for all datasets, and the total number of rounds to 5 for CIFAR-100, ImageNet-R, EuroSAT, and CUB-200. Given the increased difficulty of ImageNet-A and Cars-196, all methods are allowed 10 communication rounds on these datasets. For Out-Of-Scope, we use a pre-trained T5-small (Raffel et al., 2020). For a complete overview of the method-specific hyperparameters, refer to appendix D.

Table 1: Evaluation on CIFAR-100, ImageNet-R and ImageNet-A. Results in terms of FAA [↑]. Best results are highlighted in bold, second-best underlined.

| | CIFAR-100 | | | ImageNet-R | | | ImageNet-A | | |
|---|---|---|---|---|---|---|---|---|---|
| **Joint** | **92.75** | | | **84.02** | | | **54.64** | | |
| **Distrib.** $\beta$ | 0.5 | 0.1 | 0.05 | 0.5 | 0.1 | 0.05 | 1.0 | 0.5 | 0.2 |
| EWC | 78.46 | 72.42 | 64.51 | 58.93 | 48.15 | 43.68 | 10.86 | 10.07 | 8.89 |
| LwF | 62.87 | 55.56 | 47.09 | 54.03 | 41.02 | 46.07 | 8.89 | 8.89 | 7.90 |
| FisherAVG | 76.10 | 74.43 | 65.31 | 58.68 | 50.82 | 47.33 | 11.59 | 11.06 | 10.14 |
| RegMean | 59.80 | 45.88 | 39.08 | 61.18 | 57.00 | 55.80 | 8.56 | 6.22 | 4.34 |
| CCVR | 79.95 | 75.14 | 65.30 | 70.00 | 62.60 | 60.38 | **39.50** | 36.27 | **35.94** |
| L2P | 83.88 | 61.54 | 55.00 | 42.08 | 23.85 | 16.98 | 20.14 | 17.31 | 16.85 |
| CODA-P | 82.25 | 61.82 | 46.74 | 61.18 | 36.73 | 25.82 | 18.30 | 14.48 | 7.31 |
| FedProto | 75.79 | 70.02 | 60.55 | 58.52 | 47.30 | 52.93 | 9.87 | 9.22 | 10.01 |
| TARGET | 74.72 | 72.32 | 62.60 | 54.65 | 45.83 | 41.32 | 10.27 | 11.39 | 10.73 |
| PILoRA | 76.48 | 75.81 | 74.80 | 53.67 | 51.62 | 49.37 | 19.62 | 18.70 | 20.01 |
| **LoRM** (ours) | **86.95** | **81.75** | **82.76** | **72.48** | **63.83** | **66.45** | 37.26 | **36.34** | 33.11 |

## 4.2 RESULTS

In Tables 1 to 3, we present the results of the evaluated approaches in terms of Final Average Accuracy (FAA). For a formal definition of this metric, please refer to appendix B.7. The results are averaged over 3 runs with different seeds, with standard deviations provided in appendix C.

In the first two in-domain datasets (Table 1), Continual Learning techniques such as LwF and EWC exhibit low to moderate performance in scenarios with limited data heterogeneity. Similarly, prompt-based CL approaches suffer from rapid performance degradation as the distribution parameter $\beta$ decreases, whereas L2P maintains strong performance at $\beta = 0.5$, ranking second overall. Model merging and FL methodologies generally underperform compared to CL approaches on CIFAR-100 but achieve better results on ImageNet-R. Among these, CCVR stands out as the top performer, attaining the second-best accuracy on the latter dataset. Despite being specifically designed for FCIL, TARGET and PILoRA demonstrate lower performance than the adapted CCVR, likely due to CCVR's server-side calibration with centralized synthetic data. Lastly, leveraging its closed-form merging technique, LoRM achieves state-of-the-art performance across all settings.

Due to the challenging adversarial nature of ImageNet-A, all approaches experience a significant drop in this setting, with only CCVR, PILoRA, and LoRM maintaining competitive accuracy. Among them, CCVR and LoRM perform comparably on average, achieving the best results.

On the EuroSAT dataset (Table 2), methodologies of the same category exhibit varying performance. Among prompt-based approaches, Coda-Prompt stands out as the second-best performer, demonstrating strong generalization to out-of-domain settings, whereas L2P falls far behind. Similarly, within Continual Learning methods, EWC significantly outperforms LwF, suggesting that distillation is less effective in this context. In general, Model Merging and Federated Learning methodologies achieve comparable results, with CCVR showing a slight advantage within this group, while LoRM surpasses all other methods by a substantial margin.

In the same table, models are evaluated based on their ability to specialize in specific domains (cars and birds). In these scenarios, CL and Model Merging techniques achieve comparable overall performance, while FL methodologies demonstrate greater specialization capabilities. Notably, FCIL-targeted approaches achieve the best results, as they explicitly account for forgetting. CCVR, PILoRA, and LoRM emerge as the top-performing methods across both datasets. They achieve similar performance on CUB-200, while LoRM outperforms the others by a large margin on Cars-196.

Table 3 presents the results on the Out-Of-Scope dataset. While CCVR often ranks second in the vision domain due to its server-side rebalancing strategy, it performs the worst in this setting, likely suffering from catastrophic forgetting. TIES-Merging surpasses CCVR, benefiting from its specialized merging strategy, while the best-performing methods leverage a closed-form solution for merging linear layers. Among these, LoRM outperforms RegMean across all three distribution-imbalance settings by incorporating LoRA, achieving state-of-the-art performance.

Table 2: Evaluation on EuroSAT, Cars-196, and CUB-200. Results in terms of FAA [↑]. Best results are highlighted in bold, second-best underlined.

| | EuroSAT | | | CARS-196 | | | CUB-200 | | |
|---|---|---|---|---|---|---|---|---|---|
| **Joint** | **98.42** | | | **85.62** | | | **86.04** | | |
| **Distrib.** $\beta$ | 1.0 | 0.5 | 0.2 | 1.0 | 0.5 | 0.2 | 1.0 | 0.5 | 0.2 |
| EWC | 64.12 | 59.30 | 56.52 | 19.55 | 18.02 | 18.29 | 31.46 | 29.60 | 27.89 |
| LwF | 31.91 | 21.26 | 31.42 | 20.84 | 22.72 | 31.76 | 25.25 | 21.11 | 18.54 |
| FisherAVG | 58.84 | 59.94 | 55.86 | 26.03 | 24.60 | 21.58 | 30.45 | 28.39 | 25.06 |
| RegMean | 48.74 | 51.73 | 45.27 | 21.83 | 20.36 | 15.92 | 35.57 | 32.84 | 32.83 |
| CCVR | 64.44 | 57.93 | 62.69 | 38.99 | 37.81 | 35.31 | 62.67 | 59.48 | 56.33 |
| L2P | 40.63 | 51.78 | 45.46 | 35.49 | 31.00 | 20.01 | 56.23 | 47.31 | 38.16 |
| CODA-P | 73.38 | 69.42 | 66.69 | 28.04 | 20.83 | 14.53 | 42.53 | 37.71 | 29.19 |
| FedProto | 58.79 | 62.85 | 64.17 | 26.08 | 24.55 | 22.75 | 30.22 | 28.27 | 26.01 |
| TARGET | 52.74 | 52.74 | 45.11 | 28.65 | 27.20 | 26.13 | 39.30 | 38.40 | 34.79 |
| PILoRA | 48.35 | 32.89 | 31.22 | 37.57 | 37.92 | 36.95 | 61.11 | 60.68 | **60.39** |
| **LoRM** (ours) | **84.23** | **77.26** | **81.36** | **54.41** | **51.87** | **48.81** | **64.60** | **63.67** | 60.06 |

Table 3: Evaluation on Out-of-Scope.

| | Out-of-Scope | | |
|---|---|---|---|
| **Joint** | **95.53** | | |
| **Distrib.** $\beta$ | 1.0 | 0.5 | 0.2 |
| CCVR | 39.18 | 38.13 | 30.36 |
| TIES | 58.84 | 54.13 | 47.71 |
| RegMean | 74.16 | 74.80 | 70.78 |
| **LoRM** (ours) | **84.78** | **82.58** | **77.67** |

Table 4: Distinct LoRM components.

| | ImageNet-R | EuroSAT |
|---|---|---|
| **Distrib.** $\beta$ | 0.5 | 1.0 |
| FedAvg | 57.65 | 41.04 |
| FedAvg *w/* LoRA | 57.00 | 53.14 |
| RegMean | 61.18 | 48.74 |
| LoRM *w/o* Eq. 9 | 75.08 | 71.40 |
| LoRM | 75.62 | 77.37 |

## 4.3 ABLATION STUDIES

*Alternating vs. Only B.* We investigate whether training the $B$ matrix only (*i.e.*, following the same procedure as LoRM but applying solely Equation 7 to merge $B$ matrices) could lead to improved performance compared to our alternating optimization approach. We conduct an experiment on EuroSAT with $\beta = 1.0$ (see Figure 3), showing that the *alternating* strategy outperforms the *only B* approach. This result underscores the importance of leveraging the entire LoRA representation capabilities to achieve superior performance.

*Rate of convergence.* Figure 2 illustrates the convergence rates of LoRM and the FCIL competitors on the first task of ImageNet-R with $\beta = 0.05$. The steeper slope of the curve for LoRM, compared to that of Target and PILoRA, showcases the faster convergence of our approach. This behavior stems from the closed-form solution applied for merging LoRA at each communication round, which accelerates convergence by leveraging its optimality. In contrast, PILoRA applies simple averaging of LoRA modules, while Target does not utilize PEFT modules at all. Additionally, the slower convergence of PILoRA compared to Target can be attributed to its prototype-based classification procedure, which necessitates a higher number of training rounds to achieve well-refined prototypes.

*LoRM components.* To evaluate the contribution of each component to the overall performance of LoRM, we conduct an ablation study by incrementally introducing each part and comparing it with baseline methods. These experiments are performed on EuroSAT with $\beta = 1.0$ and ImageNet-R with $\beta = 0.5$. The initial baseline for both datasets is FedAvg (McMahan et al., 2017) combined with ACE (Caccia et al., 2022). Notably, leveraging LoRA leads to a remarkable performance improvement on EuroSAT but does not affect ImageNet-R. Meanwhile, using the closed-form aggregation of RegMean, enhanced with ACE, establishes an additional baseline. Building upon this, our alternating optimization strategy, when applied without Equation 9 (*i.e.*, simply averaging task-specific residuals), substantially boosts performance on both datasets, with a more pronounced impact on the out-of-domain dataset. Finally, applying RegMean to merge task-specific residuals further enhances robustness in addressing FCIL challenges, resulting in the final form of LoRM.

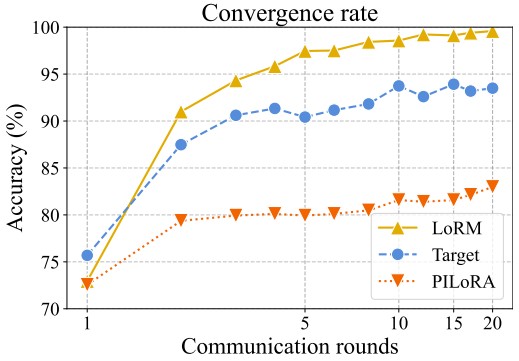

Figure 2: Speed of convergence.

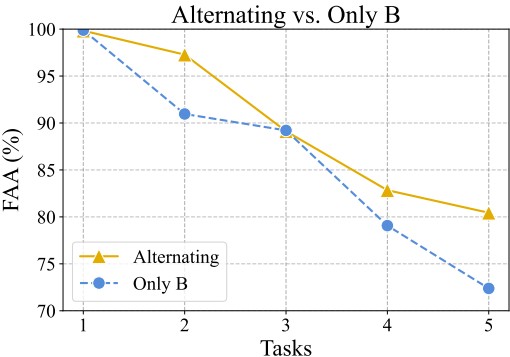

Figure 3: Alternating optimization.

## 5 RELATION WITH PRIOR WORKS

### PARAMETER-EFFICIENT FINE-TUNING

Adapting deep neural networks to new tasks typically requires full re-training of all parameters, which is computationally demanding. Parameter-Efficient Fine-Tuning (PEFT) addresses this by updating only a small subset of the model parameters, leaving the rest of the network unchanged. One of the earliest methods in this domain is Adapters (Houlsby et al., 2019), which are lightweight neural modules inserted between the layers of a pre-trained model to facilitate task adaptation. More recently, a prominent class of PEFT techniques includes Prompt Tuning (Lester et al., 2021), Prefix Tuning (Li & Liang, 2021), and Context Optimization (Zhou et al., 2022); they all introduce learnable embeddings, or prompts, appended to the layers' input tokens. Specifically, Prompt Tuning prepends these embeddings directly to the initial input sequence, whereas Prefix Tuning concatenates them to specific attention layers, and context optimization learns textual prompts in Vision-Language models. Recent adaptation for continual learning include L2P (Wang et al., 2022b), CODA-Prompt (Smith et al., 2023), and CGIL (Frascaroli et al., 2024). Alternatively, Low-Rank Adaptation (LoRA) (Hu et al., 2022) injects residual parameters into the pre-trained weights using low-rank matrices, significantly reducing the number of learnable parameters. While recent works have explored various alternatives to LoRA(Kopiczko et al., 2024; Zhang et al., 2023d; Renduchintala et al., 2024; Liu et al., 2022), our research is grounded in the original LoRA framework, leveraging its efficiency and the extensive literature on model merging that goes with it.

### MODEL MERGING

Task-specific modules are typically deployed to address individual tasks. However, an alternative approach involves merging these modules to create a model capable of generalizing across all tasks. A notable line of research within model merging focuses on combining task vectors (Ilharco et al., 2023), specifically those generated by LoRA-like methods. Early studies have explored the use of linear arithmetic operations (*e.g.*, addition and subtraction) to combine these modules (Zhang et al., 2023c). Expanding on this, subsequent works have focused on optimizing the coefficients that determine the contribution of each LoRA module during aggregation (Huang et al., 2024; Yang et al., 2024; Yadav et al., 2024; Wu et al., 2024). Alternatively, RegMean (Jin et al., 2023) introduces a closed-form solution for merging model weights, though its application to LoRA module merging has not yet been explored. In this work, we present a novel methodology that leverages this closed-form solution, specifically tailored for the Federated Class-Incremental Learning setting.

### FEDERATED CLASS-INCREMENTAL LEARNING

Federated Class-Incremental Learning (FCIL), introduced by Yoon et al. (2021), addresses the challenges inherent in Continual Learning, such as Catastrophic Forgetting (Robins, 1995), and Federated Learning, such as Client Drift (Zhao et al., 2018; Karimireddy et al., 2020), within a unified framework. Although the FCIL literature is still in its early stages, several studies have adapted popular Continual Learning methods to the federated paradigm, often by employing the

FedAvg (McMahan et al., 2017) technique for aggregating client model weights. A few approaches, however, are specifically designed for the FCIL setting. For instance, GLFC (Dong et al., 2022) incorporates local buffers and a regularization technique to mitigate the effects of gradient updates related to novel classes, with further refinements in a subsequent work by the same authors (Dong et al., 2023). TARGET (Zhang et al., 2023b) trains a centralized GAN (Goodfellow et al., 2014) to populate replay buffers with synthetic samples. Fed-CPrompt (Bagwe et al., 2023) combines local and global prompts to address both Continual Learning and Federated Learning simultaneously. Recently, HGP (Salami et al., 2024) and PILoRA (Guo et al., 2024) have integrated PEFT techniques with prototypes to mitigate forgetting and client drift. While PILoRA combines LoRA with prototypes for classification, our proposed methodology leverages LoRA modules and aggregates them by aligning their output in both Continual and Federated scenarios in a closed-form.

# 6 Conclusions

At the outset of our research, we explored the possibility of a closed-form solution for merging LoRA modules. Motivated by this question, we identified and validated the existence of such a solution, demonstrating its practicality within the framework of Federated Class-Incremental Learning (FCIL). We introduced LoRM, a method specifically designed for the FCIL scenario, which sets a new State of the Art in this domain.

Looking forward, we aim to broaden our exploration to encompass a wider array of PEFT modules. For modules such as VeRA (Kopiczko et al., 2024) and (IA)$^3$ (Liu et al., 2022), we have already derived closed-form merging equations, which are detailed in appendices A.2 and A.3, respectively. Furthermore, given the generality of our derivations beyond the FCIL setting, we also intend to evaluate the robustness of our equations across various model merging scenarios. Ultimately, we hope this work will serve not only the Federated Learning and Continual Learning communities but also contribute to the growing body of research on efficient module compositionally.

## Acknowledgments

This article is the result of a collaboration with the partners of the STORE project (*https://edf-store.com*), funded by the European Defence Fund (EDF) under grant agreement EDF-2022-101121405-STORE.

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

SUPPLEMENTARY MATERIAL OVERVIEW

This supplementary document provides detailed mathematical derivations and additional information to complement the main text. The contents are organized as follows:

- *Section A – Mathematical Derivations:* presents the mathematical derivations related to the combination of different Parameter-Efficient Fine-Tuning methods with RegMean.
  - *Section A.1 – Combination of LoRA and RegMean:* derives the optimization problem when integrating Low-Rank Adaptation (LoRA) with RegMean.
    * *Section A.1.1 – Solving for $A$ by Fixing $B$:* details the solution for matrix $A$ when matrix $B$ is held constant.
    * *Section A.1.2 – Solving for $B$ by Fixing $A$:* details the solution for matrix $B$ when matrix $A$ is held constant.
  - *Section A.2 – Combination of VeRA and RegMean:* extends the derivations to the Vector-based Random matrix Adaptation (VeRA) method.
  - *Section A.3 – Combination of $(IA)^3$ and RegMean:* explores how the $(IA)^3$ method can be combined with RegMean and provides the corresponding derivations.
  - *Section A.4 – RegMean Applied to Classification Heads:* discusses the application of RegMean to classification heads in a class-incremental learning setting.
- *Section B – Details on Datasets and Metrics:* offers comprehensive details about the datasets used and the evaluation metric employed.
- *Section C – Standard Deviations:* presents the standard deviations for all experimental results.
- *Section D – Hyperparameters:* details the hyperparameters used across all experiments.

# A    MATHEMATICAL DERIVATIONS

## A.1    COMBINATION OF LoRA AND REGMEAN

We consider a minimization problem over a single linear layer where $X_i$ represents the input to the layer. The objective function is defined as:

$$\text{minimize } \Omega = \sum_{i=1}^{N} \|WX_i - W_iX_i\|_2^2. \tag{10}$$

Here, $W$ and $W_i$ are the weight matrices for the global model (server) and the local models (clients) respectively, while $N$ represents the number of clients. In the context of a LoRA residual module, we replace $W$ with $W_0 + BA$, where $W_0$ denotes the pre-trained weight matrix and $BA$ represents the learned low-rank difference. The objective function becomes:

$$\Omega = \sum_{i=1}^{N} \|(W_0 + BA)X_i - (W_0 + B_iA_i)X_i\|_2^2.$$

Expanding the previous expression yields:

$$\Omega = \sum_{i=1}^{N} \|W_0X_i - W_0X_i + BAX_i - B_iA_iX_i\|_2^2.$$

Which simplifies the equation to:

$$\Omega = \sum_{i=1}^{N} \|BAX_i - B_iA_iX_i\|_2^2.$$

It is important to note that $B$ and $A$ refer to the merged LoRA matrices ($B_M$ and $A_M$ in the main paper). For simplicity, we omit the subscript $_M$. To minimize this objective function, we compute the partial derivatives with respect to the unknown matrices $A$ and $B$, setting them equal to zero. This leads to the following system of equations:

$$\begin{cases} \dfrac{\partial \Omega}{\partial A} = 0 & \Rightarrow \quad \sum_{i=1}^{N} 2B^\top(BAX_i - B_iA_iX_i)X_i^\top = 0 \\[2mm] \dfrac{\partial \Omega}{\partial B} = 0 & \Rightarrow \quad \sum_{i=1}^{N} 2(BAX_i - B_iA_iX_i)X_i^\top A^\top = 0 \end{cases}$$

We can rearrange the equations as:

$$\begin{cases} \sum_{i=1}^{N} B^\top BAX_iX_i^\top = \sum_{i=1}^{N} B^\top B_iA_iX_iX_i^\top \\[2mm] \sum_{i=1}^{N} BAX_iX_i^\top A^\top = \sum_{i=1}^{N} B_iA_iX_iX_i^\top A^\top \end{cases}$$

Next, we multiply the second equation on the left by $B^\top$, in order to align it with the structure of the first equation:

$$\sum_{i=1}^{N} B^\top B_iA_iX_iX_i^\top A^\top = \sum_{i=1}^{N} B^\top B_iA_iX_iX_i^\top A^\top.$$

By substituting the left-hand side of the first equation into the second, we obtain:

$$\sum_{i=1}^{N} B^\top B_iA_iX_iX_i^\top A^\top - \sum_{i=1}^{N} B^\top B_iA_iX_iX_i^\top A^\top = 0.$$

This simplifies to $0 = 0$, indicating that the system is indeterminate, with infinitely many possible solutions.

### A.1.1 Solving for $A$ by Fixing $B$

Assuming the matrix $\boldsymbol{B}$ is fixed, we can solve for $\boldsymbol{A}$. The objective function $\Omega$ becomes:

$$\Omega_{\boldsymbol{A}} = \sum_{i=1}^{N} \|\boldsymbol{B}\boldsymbol{A}\boldsymbol{X}_i - \boldsymbol{B}\boldsymbol{A}_i\boldsymbol{X}_i\|_2^2.$$

To minimize $\Omega_{\boldsymbol{A}}$, we set its derivative with respect to $\boldsymbol{A}$ to zero:

$$\frac{\partial \Omega_{\boldsymbol{A}}}{\partial \boldsymbol{A}} = 0 \quad \Rightarrow \quad \sum_{i=1}^{N} 2\boldsymbol{B}^\top(\boldsymbol{B}\boldsymbol{A}\boldsymbol{X}_i - \boldsymbol{B}\boldsymbol{A}_i\boldsymbol{X}_i)\boldsymbol{X}_i^\top = 0,$$

which simplifies to:

$$\sum_{i=1}^{N} \boldsymbol{B}^\top\boldsymbol{B}(\boldsymbol{A}\boldsymbol{X}_i - \boldsymbol{A}_i\boldsymbol{X}_i)\boldsymbol{X}_i^\top = 0.$$

Since $\boldsymbol{B}\boldsymbol{A}$ is designed to be low-rank, the matrix $\boldsymbol{B}$ has more rows than columns. As a result, provided that no row or column of $\boldsymbol{B}$ is zero, the inverse $(\boldsymbol{B}^\top\boldsymbol{B})^{-1}$ exists. By multiplying both sides of the equation from the left by $(\boldsymbol{B}^\top\boldsymbol{B})^{-1}$, we obtain:

$$\sum_{i=1}^{N}(\boldsymbol{A}\boldsymbol{X}_i - \boldsymbol{A}_i\boldsymbol{X}_i)\boldsymbol{X}_i^\top = 0, \quad \Rightarrow \quad \boldsymbol{A}\sum_{i=1}^{N}\boldsymbol{X}_i\boldsymbol{X}_i^\top = \sum_{i=1}^{N}\boldsymbol{A}_i\boldsymbol{X}_i\boldsymbol{X}_i^\top.$$

Finally, solving for $\boldsymbol{A}$, we get:

$$\boldsymbol{A} = \left(\sum_{i=1}^{N}\boldsymbol{A}_i\boldsymbol{X}_i\boldsymbol{X}_i^\top\right)\left(\sum_{i=1}^{N}\boldsymbol{X}_i\boldsymbol{X}_i^\top\right)^{-1}.$$

### A.1.2 Solving for $B$ by Fixing $A$

When the matrix $\boldsymbol{A}$ is fixed, we can solve for $\boldsymbol{B}$. The objective function $\Omega$ becomes:

$$\Omega_{\boldsymbol{B}} = \sum_{i=1}^{N} \|\boldsymbol{B}\boldsymbol{A}\boldsymbol{X}_i - \boldsymbol{B}_i\boldsymbol{A}\boldsymbol{X}_i\|_2^2.$$

To minimize $\Omega_{\boldsymbol{B}}$, we set its derivative with respect to $\boldsymbol{B}$ equal to zero:

$$\frac{\partial \Omega_{\boldsymbol{B}}}{\partial \boldsymbol{B}} = 0 \quad \Rightarrow \quad \sum_{i=1}^{N} 2(\boldsymbol{B}\boldsymbol{A}\boldsymbol{X}_i - \boldsymbol{B}_i\boldsymbol{A}\boldsymbol{X}_i)\boldsymbol{X}_i^\top\boldsymbol{A}^\top = 0.$$

Expanding the equation, we get:

$$\sum_{i=1}^{N}\boldsymbol{B}\boldsymbol{A}\boldsymbol{X}_i\boldsymbol{X}_i^\top\boldsymbol{A}^\top - \sum_{i=1}^{N}\boldsymbol{B}_i\boldsymbol{A}\boldsymbol{X}_i\boldsymbol{X}_i^\top\boldsymbol{A}^\top = 0,$$

which simplifies to:

$$\boldsymbol{B}\sum_{i=1}^{N}\boldsymbol{A}\boldsymbol{X}_i\boldsymbol{X}_i^\top\boldsymbol{A}^\top = \sum_{i=1}^{N}\boldsymbol{B}_i\boldsymbol{A}\boldsymbol{X}_i\boldsymbol{X}_i^\top\boldsymbol{A}^\top.$$

Finally, solving for $\boldsymbol{B}$, we obtain:

$$\boldsymbol{B} = \left(\sum_{i=1}^{N}\boldsymbol{B}_i\boldsymbol{A}\boldsymbol{X}_i\boldsymbol{X}_i^\top\right)\boldsymbol{A}^\top\left(\sum_{i=1}^{N}\boldsymbol{A}\boldsymbol{X}_i\boldsymbol{X}_i^\top\boldsymbol{A}^\top\right)^{-1}.$$

## A.2 COMBINATION OF VERA AND REGMEAN

Following appendix A.1, we consider the same minimization problem as Equation 10. In the context of a VeRA residual module, we replace $\boldsymbol{W}$ with $\boldsymbol{W}_0 + \Lambda_b \boldsymbol{B} \Lambda_d \boldsymbol{A}$, where $\boldsymbol{W}_0$ denotes the pre-trained weight matrix and $\Lambda_b \boldsymbol{B} \Lambda_d \boldsymbol{A}$ represents the learned weight difference. More precisely, $\boldsymbol{B}$ and $\boldsymbol{A}$ are randomly initialized, while $\Lambda_b$ and $\Lambda_d$ are learned. The objective function becomes:

$$\Omega = \sum_{i=1}^{N} \|(\boldsymbol{W}_0 + \Lambda_b \boldsymbol{B} \Lambda_d \boldsymbol{A})\boldsymbol{X}_i - (\boldsymbol{W}_0 + \Lambda_{b,i} \boldsymbol{B} \Lambda_{d,i} \boldsymbol{A})\boldsymbol{X}_i\|_2^2.$$

While every component of LoRA is a full matrix, here we are considering $\Lambda_b$ and $\Lambda_d$, which are diagonal matrices. In order to enforce this constraint into the previous formulation, we can rewrite them as column vectors $\lambda^b$ and $\lambda^d$:

$$\Omega = \sum_{i=1}^{N} \left\| \left(\boldsymbol{W}_0 + \left((\lambda^b \boldsymbol{1}_b) \odot \boldsymbol{B}\right) \left((\lambda^d \boldsymbol{1}_d) \odot \boldsymbol{A}\right)\right) \boldsymbol{X}_i - (\boldsymbol{W}_0 + \left((\lambda_i^b \boldsymbol{1}_b) \odot \boldsymbol{B}\right) \left((\lambda_i^d \boldsymbol{1}_d) \odot \boldsymbol{A}\right))\boldsymbol{X}_i \right\|_2^2,$$

where $\boldsymbol{1}_b$ and $\boldsymbol{1}_d$ are two row vectors of ones with the same columns as $\boldsymbol{B}$ and $\boldsymbol{A}$, respectively. Please note that in our notation, indices of column vectors are denoted as superscripts, while those of row vectors are represented as subscripts. Expanding this objective, we can write:

$$\Omega = \sum_{i=1}^{N} \|\boldsymbol{W}_0 \boldsymbol{X}_i - \boldsymbol{W}_0 \boldsymbol{X}_i + \left((\lambda^b \boldsymbol{1}_b) \odot \boldsymbol{B}\right) \left((\lambda^d \boldsymbol{1}_d) \odot \boldsymbol{A}\right) \boldsymbol{X}_i - \left((\lambda_i^b \boldsymbol{1}_d) \odot \boldsymbol{B}\right) \left((\lambda_i^d \boldsymbol{1}_d) \odot \boldsymbol{A}\right) \boldsymbol{X}_i\|_2^2.$$

Which simplifies the equation to:

$$\Omega = \sum_{i=1}^{N} \|\left((\lambda^b \boldsymbol{1}_b) \odot \boldsymbol{B}\right) \left((\lambda^d \boldsymbol{1}_d) \odot \boldsymbol{A}\right) \boldsymbol{X}_i - \left((\lambda_i^b \boldsymbol{1}_d) \odot \boldsymbol{B}\right) \left((\lambda_i^d \boldsymbol{1}_d) \odot \boldsymbol{A}\right) \boldsymbol{X}_i\|_2^2.$$

If we compute the partial derivatives with respect to the unknown vectors $\lambda^b$ and $\lambda^d$ and set them equal to zero, we obtain an indeterminate system, as similarly discussed in appendix A.1. The detailed derivation is left to the reader. However, it is still possible to solve for either $\lambda^b$ or $\lambda^d$ when the other is known.

### A.2.1 SOLVING FOR $\lambda^d$ WITH FIXED $\lambda^b$

Assuming the vector $\lambda^b$ is fixed, we can solve for $\lambda^d$. The objective function $\Omega$ becomes:

$$\Omega_{\lambda^d} = \sum_{i=1}^{N} \left\| \left((\lambda^b \boldsymbol{1}_b) \odot \boldsymbol{B}\right) \left((\lambda^d \boldsymbol{1}_d) \odot \boldsymbol{A}\right) \boldsymbol{X}_i - \left((\lambda^b \boldsymbol{1}_b) \odot \boldsymbol{B}\right) \left((\lambda_i^d \boldsymbol{1}_d) \odot \boldsymbol{A}\right) \boldsymbol{X}_i \right\|_2^2.$$

This can be rearranged as:

$$\Omega_{\lambda^d} = \sum_{i=1}^{N} \left\| \left((\lambda^b \boldsymbol{1}_b) \odot \boldsymbol{B}\right) \left[ \left((\lambda^d \boldsymbol{1}_d) \odot \boldsymbol{A}\right) \boldsymbol{X}_i - \left((\lambda_i^d \boldsymbol{1}_d) \odot \boldsymbol{A}\right) \boldsymbol{X}_i \right] \right\|_2^2.$$

Since $(\lambda^b \boldsymbol{1}_b) \odot \boldsymbol{B}$ is a multiplicative constant, it can be factored out and disregarded, leading to the following simplified objective:

$$\Omega_{\lambda^d} = \sum_{i=1}^{N} \left\| \left((\lambda^d \boldsymbol{1}_d) \odot \boldsymbol{A}\right) \boldsymbol{X}_i - \left((\lambda_i^d \boldsymbol{1}_d) \odot \boldsymbol{A}\right) \boldsymbol{X}_i \right\|_2^2.$$

To minimize $\Omega_{\lambda^d}$, we take its derivative with respect to $\lambda^d$ and set it to zero:

$$\frac{\partial \Omega_{\lambda^d}}{\partial \lambda^d} = 0 \quad \Rightarrow \quad 2\sum_{i=1}^{N} ((\lambda^d \boldsymbol{1}_d) \odot \boldsymbol{A})\boldsymbol{X}_i \boldsymbol{X}_i^\top \odot \boldsymbol{A}\boldsymbol{1}_d^\top - \sum_{i=1}^{N} ((\lambda_i^d \boldsymbol{1}_d) \odot \boldsymbol{A})\boldsymbol{X}_i \boldsymbol{X}_i^\top \odot \boldsymbol{A}\boldsymbol{1}_d^\top = 0.$$

This simplifies to:

$$((\lambda^d \boldsymbol{1}_d) \odot \boldsymbol{A}) \sum_{i=1}^{N} \boldsymbol{X}_i \boldsymbol{X}_i^\top = \sum_{i=1}^{N} ((\lambda_i^d \boldsymbol{1}_d) \odot \boldsymbol{A})\boldsymbol{X}_i \boldsymbol{X}_i^\top.$$

We then solve for $\lambda^d$ as follows:

$$((\lambda^d \mathbf{1}_d) \odot \boldsymbol{A}) = \left( \sum_{i=1}^{N} ((\lambda_i^d \mathbf{1}_d) \odot \boldsymbol{A}) \boldsymbol{X}_i \boldsymbol{X}_i^\top \right) \left( \sum_{i=1}^{N} \boldsymbol{X}_i \boldsymbol{X}_i^\top \right)^{-1},$$

$$\lambda^d \mathbf{1}_d = \left( \sum_{i=1}^{N} ((\lambda_i^d \mathbf{1}_d) \odot \boldsymbol{A}) \boldsymbol{X}_i \boldsymbol{X}_i^\top \right) \left( \sum_{i=1}^{N} \boldsymbol{X}_i \boldsymbol{X}_i^\top \right)^{-1} \odot \frac{1}{\boldsymbol{A}},$$

$$\lambda^d \mathbf{1}_d \mathbf{1}_d^\top = \left( \sum_{i=1}^{N} ((\lambda_i^d \mathbf{1}_d) \odot \boldsymbol{A}) \boldsymbol{X}_i \boldsymbol{X}_i^\top \right) \left( \sum_{i=1}^{N} \boldsymbol{X}_i \boldsymbol{X}_i^\top \right)^{-1} \odot \frac{1}{\boldsymbol{A}} \mathbf{1}_d^\top.$$

Since $\mathbf{1}_d \mathbf{1}_d^\top = d$ is a scalar, we can finally find:

$$\lambda^d = \left( \sum_{i=1}^{N} ((\lambda_i^d \mathbf{1}_d) \odot \boldsymbol{A}) \boldsymbol{X}_i \boldsymbol{X}_i^\top \right) \left( \sum_{i=1}^{N} \boldsymbol{X}_i \boldsymbol{X}_i^\top \right)^{-1} \odot \frac{1}{d\boldsymbol{A}} \mathbf{1}_d^\top.$$

### A.2.2 SOLVING FOR $\lambda^b$ WITH FIXED $\lambda^d$

When the vector $\lambda^d$ is fixed, we solve for $\lambda^b$. The objective function $\Omega$ becomes:

$$\Omega_{\lambda^b} = \sum_{i=1}^{N} \left\| \left( (\lambda^b \mathbf{1}_b) \odot \boldsymbol{B} \right) \left( (\lambda^d \mathbf{1}_d) \odot \boldsymbol{A} \right) \boldsymbol{X}_i - \left( (\lambda_i^b \mathbf{1}_b) \odot \boldsymbol{B} \right) \left( (\lambda^d \mathbf{1}_d) \odot \boldsymbol{A} \right) \boldsymbol{X}_i \right\|_2^2.$$

To minimize $\Omega_{\lambda^b}$, we compute its derivative with respect to $\lambda^b$ and set it to zero:

$$\frac{\partial \Omega_{\lambda^b}}{\partial \lambda^b} = 0 \quad \Rightarrow \quad 2 \sum_{i=1}^{N} \Big[ ((\lambda^b \mathbf{1}_b) \odot \boldsymbol{B})((\lambda^d \mathbf{1}_d) \odot \boldsymbol{A}) \boldsymbol{X}_i \boldsymbol{X}_i^\top -$$

$$- ((\lambda_i^b \mathbf{1}_b) \odot \boldsymbol{B})((\lambda^d \mathbf{1}_d) \odot \boldsymbol{A}) \boldsymbol{X}_i \boldsymbol{X}_i^\top \Big] ((\lambda^d \mathbf{1}_d) \odot \boldsymbol{A})^\top \odot \boldsymbol{B} \mathbf{1}_b^\top = 0.$$

This simplifies to:

$$((\lambda^b \mathbf{1}_b) \odot \boldsymbol{B}) \sum_{i=1}^{N} \mathcal{A} \boldsymbol{X}_i \boldsymbol{X}_i^\top \mathcal{A}^\top = \sum_{i=1}^{N} ((\lambda_i^b \mathbf{1}_b) \odot \boldsymbol{B}) \mathcal{A} \boldsymbol{X}_i \boldsymbol{X}_i^\top \mathcal{A}^\top,$$

where we use $\mathcal{A}$ to denote $((\lambda^d \mathbf{1}_d) \odot \boldsymbol{A})$ for simplicity. Solving for $\lambda^b$, we get:

$$((\lambda^b \mathbf{1}_b) \odot \boldsymbol{B}) = \left( \sum_{i=1}^{N} ((\lambda_i^b \mathbf{1}_b) \odot \boldsymbol{B}) \mathcal{A} \boldsymbol{X}_i \boldsymbol{X}_i^\top \mathcal{A}^\top \right) \left( \sum_{i=1}^{N} \mathcal{A} \boldsymbol{X}_i \boldsymbol{X}_i^\top \mathcal{A}^\top \right)^{-1},$$

$$\lambda^b \mathbf{1}_b = \left( \sum_{i=1}^{N} ((\lambda_i^b \mathbf{1}_b) \odot \boldsymbol{B}) \mathcal{A} \boldsymbol{X}_i \boldsymbol{X}_i^\top \mathcal{A}^\top \right) \left( \sum_{i=1}^{N} \mathcal{A} \boldsymbol{X}_i \boldsymbol{X}_i^\top \mathcal{A}^\top \right)^{-1} \odot \frac{1}{\boldsymbol{B}},$$

$$\lambda^b \mathbf{1}_b \mathbf{1}_d^\top = \left( \sum_{i=1}^{N} ((\lambda_i^b \mathbf{1}_b) \odot \boldsymbol{B}) \mathcal{A} \boldsymbol{X}_i \boldsymbol{X}_i^\top \mathcal{A}^\top \right) \left( \sum_{i=1}^{N} \mathcal{A} \boldsymbol{X}_i \boldsymbol{X}_i^\top \mathcal{A}^\top \right)^{-1} \odot \frac{1}{\boldsymbol{B}} \mathbf{1}_d^\top.$$

Finally, since $\mathbf{1}_d \mathbf{1}_d^\top = d$ is a scalar, we can solve for $\lambda^b$:

$$\lambda^b = \left( \sum_{i=1}^{N} ((\lambda_i^b \mathbf{1}_b) \odot \boldsymbol{B}) \mathcal{A} \boldsymbol{X}_i \boldsymbol{X}_i^\top \mathcal{A}^\top \right) \left( \sum_{i=1}^{N} \mathcal{A} \boldsymbol{X}_i \boldsymbol{X}_i^\top \mathcal{A}^\top \right)^{-1} \odot \frac{1}{d\boldsymbol{B}} \mathbf{1}_d^\top,$$

where $\mathcal{A} = ((\lambda^d \mathbf{1}_d) \odot \boldsymbol{A})$.

## A.3 COMBINATION OF (IA)³ AND REGMEAN

(IA)³ (Liu et al., 2022) adapts the pretrained model by learning multiplicative weight vectors $\ell^k$ that modulate the intermediate activations — where $k$ is the dimension of the considered activation. Specifically, for each attention layer, these weight vectors are applied to the input keys, values, and output. Notably, it can be demonstrated that, for a single linear layer, (IA)³ can be expressed as a residual module (Porrello et al., 2024):

$$\ell^k \odot (\boldsymbol{W}_0 \boldsymbol{X}) = \left[ \boldsymbol{W}_0 + ((\ell^k - \mathbf{1}_k^\top)\mathbf{1}_h) \odot \boldsymbol{W}_0 \right] \boldsymbol{X},$$

where $\mathbf{1}_k$ and $\mathbf{1}_h$ represent row vectors of ones, respectively corresponding to the dimensions of the output and input of the considered layer. For the sake of simplicity, we deviate from the original formulation, which initializes $\ell^k$ as a column vector of ones. Instead, we begin with a column vector of zeroes at the beginning of training. This adjustment does not affect the outcome, as we simultaneously modify our formulation as follows:

$$\ell^k \odot (\boldsymbol{W}_0 \boldsymbol{X}) = \left[ \boldsymbol{W}_0 + (\ell^k \mathbf{1}_h) \odot \boldsymbol{W}_0 \right] \boldsymbol{X},$$

Then, following appendix A.1, we consider the same minimization problem as Equation 10. This time, we replace $\boldsymbol{W}$ with $\boldsymbol{W}_0 + (\ell^k \mathbf{1}_h) \odot \boldsymbol{W}_0$, where $\boldsymbol{W}_0$ denotes the pre-trained weight matrix and $(\ell^k \mathbf{1}_h) \odot \boldsymbol{W}_0$ represents the learned weight difference and $\ell^k$ is zero initialized. The objective function becomes:

$$\Omega_{(\text{IA})^3} = \sum_{i=1}^{N} \|(\boldsymbol{W}_0 + (\ell^k \mathbf{1}_h) \odot \boldsymbol{W}_0)\boldsymbol{X}_i - (\boldsymbol{W}_0 + (\ell_i^k \mathbf{1}_h) \odot \boldsymbol{W}_0)\boldsymbol{X}_i\|_2^2.$$

Following appendix A.1, we can rearrange $\Omega_{(\text{IA})^3}$ as:

$$\Omega_{(\text{IA})^3} = \sum_{i=1}^{N} \|((\ell^k \mathbf{1}_h) \odot \boldsymbol{W}_0)\boldsymbol{X}_i - ((\ell_i^k \mathbf{1}_h) \odot \boldsymbol{W}_0)\boldsymbol{X}_i\|_2^2.$$

To minimize $\Omega_{(\text{IA})^3}$, we take its derivative with respect to $\ell^k$ and set it to zero:

$$\frac{\partial \Omega_{(\text{IA})^3}}{\partial \ell^k} = 0 \quad \Rightarrow \quad 2\sum_{i=1}^{N} \left[ ((\ell^k \mathbf{1}_h) \odot \boldsymbol{W}_0)\boldsymbol{X}_i \boldsymbol{X}_i^\top - ((\ell_i^k \mathbf{1}_h) \odot \boldsymbol{W}_0)\boldsymbol{X}_i \boldsymbol{X}_i^\top \right] \odot \boldsymbol{W}_0 \mathbf{1}_h^\top = 0.$$

This simplifies to:

$$((\ell^k \mathbf{1}_h) \odot \boldsymbol{W}_0) \sum_{i=1}^{N} \boldsymbol{X}_i \boldsymbol{X}_i^\top = \sum_{i=1}^{N} ((\ell_i^k \mathbf{1}_h) \odot \boldsymbol{W}_0)\boldsymbol{X}_i \boldsymbol{X}_i^\top.$$

Solving for $\ell^k$, we get:

$$((\ell^k \mathbf{1}_h) \odot \boldsymbol{W}_0) = \left( \sum_{i=1}^{N} ((\ell_i^k \mathbf{1}_h) \odot \boldsymbol{W}_0)\boldsymbol{X}_i \boldsymbol{X}_i^\top \right) \left( \sum_{i=1}^{N} \boldsymbol{X}_i \boldsymbol{X}_i^\top \right)^{-1}.$$

$$\ell^k \mathbf{1}_h = \left( \sum_{i=1}^{N} ((\ell_i^k \mathbf{1}_h) \odot \boldsymbol{W}_0)\boldsymbol{X}_i \boldsymbol{X}_i^\top \right) \left( \sum_{i=1}^{N} \boldsymbol{X}_i \boldsymbol{X}_i^\top \right)^{-1} \odot \frac{1}{\boldsymbol{W}_0}.$$

$$\ell^k \mathbf{1}_h \mathbf{1}_h^\top = \left( \sum_{i=1}^{N} ((\ell_i^k \mathbf{1}_h) \odot \boldsymbol{W}_0)\boldsymbol{X}_i \boldsymbol{X}_i^\top \right) \left( \sum_{i=1}^{N} \boldsymbol{X}_i \boldsymbol{X}_i^\top \right)^{-1} \odot \frac{1}{\boldsymbol{W}_0} \mathbf{1}_h^\top.$$

Finally, since $\mathbf{1}_h \mathbf{1}_h^\top = h$ is a scalar, we can solve for $\ell^k$:

$$\ell^k = \left( \sum_{i=1}^{N} ((\ell_i^k \mathbf{1}_h) \odot \boldsymbol{W}_0)\boldsymbol{X}_i \boldsymbol{X}_i^\top \right) \left( \sum_{i=1}^{N} \boldsymbol{X}_i \boldsymbol{X}_i^\top \right)^{-1} \odot \frac{1}{h\boldsymbol{W}_0} \mathbf{1}_h^\top.$$

### A.4 REGMEAN APPLIED TO CLASSIFICATION HEADS

While each client adapts the pretrained model to the specific task using residual parameter-efficient modules, the classification head must be re-initialized and trained from scratch. Specifically, following the approach of Caccia et al. (2022), we initialize a distinct classification head for each task. This practice has recently emerged as the *de facto* standard in Class-Incremental Learning, consistently providing a performance boost across various methodologies.

In the context of Class-Incremental Learning, all task-specific classification heads must eventually be merged into a unified classifier. To address this, we aim to minimize the RegMean problem, as defined in Equation 10, across the classifiers of different tasks. Rather than considering a separate set of parameters for each client, we now introduce a weight matrix for each task:

$$\text{minimize } \Omega_{\text{cls}} = \sum_{t=1}^{T} \|\boldsymbol{W}\boldsymbol{X}_t - \boldsymbol{W}_t\boldsymbol{X}_t\|_2^2.$$

Notably, in this case, $\boldsymbol{W}$ and $\boldsymbol{W}_t$ differ in dimensionality, as the classification heads $\{\boldsymbol{W}_1, \boldsymbol{W}_2, \ldots, \boldsymbol{W}_T\}$ are trained independently. Specifically, while $\boldsymbol{W}$ and each $\boldsymbol{W}_t$ share the same number of columns (input dimension), they differ in the number of rows (output dimension). The total number of rows of $\boldsymbol{W}$ corresponds to the sum of the rows of all $\boldsymbol{W}_t$, which is equivalent to the total number of classes $\mathcal{C}$; instead, each $\boldsymbol{W}_t$ has $c$ rows. This structure enables us to decompose $\Omega_{\text{cls}}$ into $T$ separate minimization problems:

$$\text{minimize } \sum_{t=1}^{T} \Omega_{\text{cls}}^t, \quad \text{where} \quad \Omega_{\text{cls}}^t = \|\boldsymbol{W}_{ct:c(t+1)}\boldsymbol{X}_t - \boldsymbol{W}_t\boldsymbol{X}_t\|_2^2.$$

Since the tasks are class-disjoint, each $\{\Omega_{\text{cls}}^1, \Omega_{\text{cls}}^2, \ldots, \Omega_{\text{cls}}^T\}$ operates on distinct rows of $\boldsymbol{W}$. Consequently, the optimal solution for each $\Omega_{\text{cls}}^t$ is to assign $\boldsymbol{W}_{ct:c(t+1)} = \boldsymbol{W}_t$ for each $t$.

## B DETAILS ON DATASETS AND METRICS

### B.1 CIFAR-100

For the incremental split of CIFAR-100, we follow the standard partitioning into 10 tasks, each consisting of 10 classes, as commonly employed in numerous Continual Learning studies (Wang et al., 2022b;a; Smith et al., 2023). For data augmentation on the training set, we apply bicubic interpolation to resize each image from $32 \times 32$ to $224 \times 224$, followed by random horizontal flipping and normalization. For the test set, each image is first resized to $256 \times 256$ using bicubic interpolation, followed by a center crop to $224 \times 224$, and a final normalization step.

### B.2 IMAGENET-R

For the incremental split of ImageNet-R, we employ the widely used partitioning into 10 tasks, each comprising 20 classes, as is common in various Continual Learning studies (Wang et al., 2022a; Smith et al., 2023; Zhang et al., 2023a). For data augmentation on the training set, we process each $224 \times 224$ image with a random horizontal flip, followed by normalization. In the test set, we first resize each image to $256 \times 256$ using bicubic interpolation, apply a center crop to $224 \times 224$, and then perform normalization.

### B.3 IMAGENET-A

For the incremental split of ImageNet-A, we partition the dataset into 10 tasks, each containing an equal number of classes. During training, augmentation includes random resized cropping to $224 \times 224$ with a scale range of $(0.05, 1.0)$ and an aspect ratio range of $(^3/_4, ^4/_3)$, followed by random horizontal flipping. Normalization is performed using a mean of $(0.485, 0.456, 0.406)$ and a standard deviation of $(0.229, 0.224, 0.225)$. The test set undergoes resizing to $256 \times 256$ using bicubic interpolation, followed by a center crop to $224 \times 224$ and the same normalization step.

### B.4 EUROSAT

For the incremental split of EuroSAT, we follow a partitioning scheme of 5 tasks, each consisting of 2 classes, consistent with prior Continual Learning studies (Jung et al., 2023; Menabue et al., 2024). No additional data augmentation or pre-processing is applied to either the training or test sets.

### B.5 CARS-196

For the incremental split of Cars-196, we divide the dataset into 10 tasks, each comprising an equal number of classes, except for the last task, which contains 16 classes. Data augmentation for the training set includes random horizontal flipping, followed by normalization using a mean of 0 and a standard deviation of 1 for all channels. The test set undergoes only the normalization step.

### B.6 CUB-200

For the incremental split of CUB-200, we follow a partitioning into 10 tasks, each containing an equal number of classes. Data augmentation for the training set includes bicubic interpolation resizing to $256 \times 256$, random cropping to $224 \times 224$, and random horizontal flipping. Normalization is applied using a mean of $(0.485, 0.456, 0.406)$ and a standard deviation of $(0.229, 0.224, 0.225)$. For the test set, each image is resized to $256 \times 256$ using bicubic interpolation, followed by a center crop to $224 \times 224$ and the same normalization step.

### B.7 FINAL AVERAGE ACCURACY (FAA)

We evaluate the performance of all methods using the Final Average Accuracy (FAA), a widely adopted metric in the FCIL literature.

FAA quantifies the mean accuracy across all tasks at the conclusion of the incremental training process. Formally, let $a^t$ represent the accuracy on the $t^{th}$ task after completing the incremental training. The FAA is defined as:

$$\text{FAA} = \frac{1}{T} \sum_{t=1}^{T} a^t, \tag{11}$$

where $T$ represents the total number of tasks.

## C STANDARD DEVIATIONS

The standard deviations for all evaluated approaches, based on three runs, are reported for the in-domain datasets in Table A and for the out-of-domain dataset in Table B.

Table A: Standard deviations (reported in FAA) for CIFAR-100, ImageNet-R and ImageNet-A.

| | CIFAR-100 | | | ImageNet-R | | | ImageNet-A | | |
|---|---|---|---|---|---|---|---|---|---|
| **Distrib. $\beta$** | 0.5 | 0.1 | 0.05 | 0.5 | 0.1 | 0.05 | 1.0 | 0.5 | 0.2 |
| EWC | 1.38 | 2.32 | 2.67 | 1.69 | 1.14 | 1.14 | 1.34 | 0.75 | 1.38 |
| LwF | 1.15 | 2.38 | 4.25 | 1.91 | 0.93 | 1.38 | 0.30 | 0.27 | 1.61 |
| FisherAVG | 0.40 | 3.67 | 3.18 | 0.77 | 0.47 | 0.93 | 1.59 | 1.36 | 1.49 |
| RegMean | 0.01 | 1.78 | 4.87 | 1.10 | 0.85 | 1.44 | 0.59 | 1.10 | 0.70 |
| CCVR | 1.13 | 2.49 | 1.39 | 0.57 | 0.49 | 2.29 | 1.19 | 0.63 | 2.26 |
| L2P | 1.62 | 1.49 | 1.85 | 1.17 | 1.61 | 1.35 | 0.72 | 0.81 | 2.01 |
| CODA-P | 1.15 | 1.96 | 0.96 | 0.98 | 1.95 | 1.19 | 0.16 | 1.63 | 0.22 |
| FedProto | 2.05 | 0.94 | 1.96 | 2.09 | 2.28 | 0.72 | 1.18 | 0.71 | 2.17 |
| TARGET | 0.79 | 1.02 | 1.81 | 0.33 | 1.58 | 1.21 | 0.65 | 0.46 | 1.34 |
| PILoRA | 0.28 | 0.98 | 4.08 | 0.29 | 1.01 | 0.33 | 0.08 | 0.31 | 0.33 |
| **LoRM** (ours) | 0.27 | 0.20 | 0.79 | 0.04 | 0.16 | 0.87 | 0.86 | 0.75 | 0.68 |

Table B: Standard deviations (reported in FAA) for EuroSAT, Cars-196, and CUB-200.

| Distrib. $\beta$ | EuroSAT | | | Cars-196 | | | CUB-200 | | |
|---|---|---|---|---|---|---|---|---|---|
| | 1.0 | 0.5 | 0.2 | 1.0 | 0.5 | 0.2 | 1.0 | 0.5 | 0.2 |
| EWC | 7.33 | 5.78 | 6.60 | 1.72 | 0.46 | 1.00 | 0.55 | 0.94 | 1.68 |
| LwF | 3.32 | 4.58 | 5.12 | 1.35 | 2.81 | 2.01 | 2.07 | 2.15 | 2.02 |
| FisherAVG | 5.43 | 6.07 | 3.40 | 2.10 | 1.78 | 0.78 | 0.26 | 1.35 | 2.21 |
| RegMean | 3.99 | 7.21 | 5.68 | 0.23 | 1.11 | 1.80 | 1.79 | 2.63 | 2.27 |
| CCVR | 9.01 | 7.16 | 6.26 | 1.49 | 0.87 | 2.08 | 1.22 | 1.69 | 1.73 |
| L2P | 2.52 | 3.49 | 2.02 | 0.87 | 1.94 | 0.36 | 2.38 | 0.78 | 1.21 |
| CODA-P | 4.52 | 6.27 | 6.54 | 1.05 | 2.24 | 1.89 | 0.51 | 1.52 | 1.86 |
| FedProto | 3.58 | 8.74 | 7.96 | 1.11 | 0.35 | 0.87 | 0.96 | 0.67 | 1.99 |
| TARGET | 4.12 | 6.20 | 5.31 | 0.93 | 0.68 | 1.54 | 1.17 | 0.65 | 2.06 |
| PILoRA | 4.45 | 4.07 | 4.27 | 0.21 | 0.33 | 0.18 | 0.51 | 0.28 | 0.51 |
| **LoRM** (ours) | 1.75 | 3.34 | 6.51 | 1.07 | 0.26 | 0.92 | 0.86 | 0.98 | 0.44 |

Table C: Standard deviations (reported in FAA) for Out-of-Scope.

| Joint | | 93.72 | |
|---|---|---|---|
| Distrib. $\beta$ | 1.0 | 0.5 | 0.2 |
| CCVR | 0.45 | 0.36 | 0.92 |
| TIES | 2.75 | 1.80 | 1.21 |
| RegMean | 0.44 | 1.30 | 1.46 |
| LoRM (ours) | **84.78** | **82.58** | **77.67** |

# D  HYPERPARAMETERS

For Method-specific hyperparameters that are not explicitly mentioned, we follow the settings described in their original papers.

## D.1  ACRONYMS

List of all the acronyms used:

- $lr$: learning rate;

- $lr_{pr}$: learning rate for prototypes;

- $\lambda_{\mathrm{KL}}$: Knowledge Distillation Loss multiplier;

- $r$: rank for low rank matrices;

- $g_{ep}$: epochs of training and generation for the generator network;

- $\gamma$: decay factor for off-diagonal elements of the Gram matrices of the backbone (first element) and of the classifier (second element).

## D.2 CIFAR-100

| Distrib. $\beta$ | 0.5 | 0.1 | 0.05 |
|---|---|---|---|
| EwC | *lr*: 1e-5 | *lr*: 1e-5 | *lr*: 1e-5 |
| LwF | *lr*: 1e-5 | *lr*: 1e-5 | *lr*: 1e-5 |
| FisherAVG | *lr*: 1e-5 | *lr*: 1e-5 | *lr*: 1e-5 |
| RegMean | *lr*: 1e-5; $\gamma$: (0.5, 0.5) | *lr*: 1e-5; $\gamma$: (0.5, 0.5) | *lr*: 1e-5; $\gamma$: (0.5, 0.5) |
| CCVR | *lr*: 1e-5 | *lr*: 1e-5 | *lr*: 1e-5 |
| L2P | *lr*: 3e-2 | *lr*: 3e-2 | *lr*: 3e-2 |
| CODA-P | *lr*: 1e-3 | *lr*: 1e-3 | *lr*: 1e-3 |
| FedProto | *lr*: 1e-5 | *lr*: 1e-5 | *lr*: 1e-5 |
| TARGET | *lr*: 1e-5; $\lambda_{\text{KL}}$: 25; $g_{ep}$: 30 | *lr*: 1e-5; $\lambda_{\text{KL}}$: 25; $g_{ep}$: 30 | *lr*: 1e-5; $\lambda_{\text{KL}}$: 25; $g_{ep}$: 30 |
| PILoRA | *lr*: 2e-2; $lr_{pr}$: 1e-4 | *lr*: 2e-2; $lr_{pr}$: 1e-4 | *lr*: 2e-2; $lr_{pr}$: 1e-4 |
| **LoRM** | *lr*: 3e-4; *r*: 1 $\gamma$: (0, 0.5) | *lr*: 1e-4; *r*: 16 $\gamma$: (0, 0.5) | *lr*: 5e-4; *r*: 16 $\gamma$: (0, 0.5) |

## D.3 IMAGENET-R

| Distrib. $\beta$ | 0.5 | 0.1 | 0.05 |
|---|---|---|---|
| EwC | *lr*: 1e-5 | *lr*: 1e-5 | *lr*: 1e-5 |
| LwF | *lr*: 1e-5 | *lr*: 1e-5 | *lr*: 3e-5 |
| FisherAVG | *lr*: 1e-5 | *lr*: 1e-5 | *lr*: 1e-5 |
| RegMean | *lr*: 1e-5; $\gamma$: (0.1, 0.1) | *lr*: 1e-5; $\gamma$: (0.1, 0.1) | *lr*: 1e-5; $\gamma$: (0.1, 0.1) |
| CCVR | *lr*: 1e-5 | *lr*: 1e-5 | *lr*: 1e-5 |
| L2P | *lr*: 3e-2 | *lr*: 3e-2 | *lr*: 3e-2 |
| CODA-P | *lr*: 1e-3 | *lr*: 1e-3 | *lr*: 1e-3 |
| FedProto | *lr*: 1e-5 | *lr*: 1e-5 | *lr*: 3e-5 |
| TARGET | *lr*: 1e-5; $\lambda_{\text{KL}}$: 25; $g_{ep}$: 30 | *lr*: 1e-5; $\lambda_{\text{KL}}$: 25; $g_{ep}$: 30 | *lr*: 1e-5; $\lambda_{\text{KL}}$: 25; $g_{ep}$: 30 |
| PILoRA | *lr*: 2e-2; $lr_{pr}$: 1e-4 | *lr*: 2e-2; $lr_{pr}$: 1e-4 | *lr*: 2e-2; $lr_{pr}$: 1e-4 |
| **LoRM** | *lr*: 3e-3; *r*: 2 $\gamma$: (0, 0.5) | *lr*: 1e-3; *r*: 32 $\gamma$: (0, 0.5) | *lr*: 1e-3; *r*: 16 $\gamma$: (0, 0.5) |

## D.4 IMAGENET-A

| Distrib. $\beta$ | 1.0 | 0.5 | 0.2 |
|---|---|---|---|
| EwC | *lr*: 1e-5 | *lr*: 1e-5 | *lr*: 1e-5 |
| LwF | *lr*: 1e-5 | *lr*: 1e-5 | *lr*: 3e-5 |
| FisherAVG | *lr*: 1e-5 | *lr*: 1e-5 | *lr*: 1e-5 |
| RegMean | *lr*: 1e-5; $\gamma$: (0.1, 0.1) | *lr*: 1e-5; $\gamma$: (0.1, 0.1) | *lr*: 1e-5; $\gamma$: (0.1, 0.1) |
| CCVR | *lr*: 1e-5 | *lr*: 1e-5 | *lr*: 1e-5 |
| L2P | *lr*: 3e-2 | *lr*: 3e-2 | *lr*: 3e-1 |
| CODA-P | *lr*: 1e-2 | *lr*: 1e-2 | *lr*: 1e-2 |
| FedProto | *lr*: 3e-5 | *lr*: 1e-5 | *lr*: 1e-5 |
| TARGET | *lr*: 1e-4; $\lambda_{\text{KL}}$: 25; $g_{ep}$: 30 | *lr*: 1e-4; $\lambda_{\text{KL}}$: 25; $g_{ep}$: 30 | *lr*: 1e-4; $\lambda_{\text{KL}}$: 25; $g_{ep}$: 30 |
| PILoRA | *lr*: 2e-2; $lr_{pr}$ 1e-4 | *lr*: 1e-2; $lr_{pr}$: 1e-4 | *lr*: 2e-2; $lr_{pr}$: 1e-4 |
| **LoRM** | *lr*: 1e-2; *r*: 4 $\gamma$: (0, 0.5) | *lr*: 1e-2; *r*: 4 $\gamma$: (0, 0.5) | *lr*: 1e-2; *r*: 4 $\gamma$: (0, 0.5) |

## D.5 EUROSAT

| Distrib. $\beta$ | 1.0 | 0.5 | 0.2 |
|---|---|---|---|
| EwC | $lr$: 1e-5 | $lr$: 1e-5 | $lr$: 1e-5 |
| LwF | $lr$: 1e-5 | $lr$: 1e-5 | $lr$: 3e-5 |
| FisherAVG | $lr$: 1e-5 | $lr$: 1e-5 | $lr$: 1e-5 |
| RegMean | $lr$: 1e-5; $\gamma$: (0.1, 0.1) | $lr$: 1e-5; $\gamma$: (0.1, 0.1) | $lr$: 1e-5; $\gamma$: (0.1, 0.1) |
| CCVR | $lr$: 1e-5 | $lr$: 1e-5 | $lr$: 1e-5 |
| L2P | $lr$: 3e-2 | $lr$: 3e-2 | $lr$: 3e-2 |
| CODA-P | $lr$: 1e-3 | $lr$: 1e-3 | $lr$: 1e-3 |
| FedProto | $lr$: 3e-5 | $lr$: 1e-5 | $lr$: 1e-5 |
| TARGET | $lr$: 1e-5; $\lambda_{\mathrm{KL}}$: 25; $g_{ep}$: 30 | $lr$: 1e-5; $\lambda_{\mathrm{KL}}$: 25; $g_{ep}$: 30 | $lr$: 1e-5; $\lambda_{\mathrm{KL}}$: 25; $g_{ep}$: 30 |
| PILoRA | $lr$: 2e-2; $lr_{pr}$: 1e-4 | $lr$: 2e-2; $lr_{pr}$: 1e-4 | $lr$: 2e-2; $lr_{pr}$: 1e-4 |
| **LoRM** | $lr$: 3e-3; $r$: 1 $\gamma$: (0, 0.5) | $lr$: 3e-3; $r$: 1 $\gamma$: (0, 0.5) | $lr$: 1e-3; $r$: 4 $\gamma$: (0, 0.5) |

## D.6 CARS-196

| Distrib. $\beta$ | 1.0 | 0.5 | 0.2 |
|---|---|---|---|
| EwC | $lr$: 1e-5 | $lr$: 1e-5 | $lr$: 1e-5 |
| LwF | $lr$: 1e-5 | $lr$: 1e-5 | $lr$: 3e-5 |
| FisherAVG | $lr$: 1e-5 | $lr$: 1e-5 | $lr$: 1e-5 |
| RegMean | $lr$: 1e-5; $\gamma$: (0.1, 0.1) | $lr$: 1e-5; $\gamma$: (0.1, 0.1) | $lr$: 1e-5; $\gamma$: (0.1, 0.1) |
| CCVR | $lr$: 1e-5 | $lr$: 1e-5 | $lr$: 1e-5 |
| L2P | $lr$: 3e-2 | $lr$: 3e-2 | $lr$: 3e-2 |
| CODA-P | $lr$: 3e-2 | $lr$: 3e-2 | $lr$: 3e-2 |
| FedProto | $lr$: 1e-5 | $lr$: 1e-5 | $lr$: 1e-5 |
| TARGET | $lr$: 1e-4; $\lambda_{\mathrm{KL}}$: 25; $g_{ep}$: 30 | $lr$: 1e-4; $\lambda_{\mathrm{KL}}$: 25; $g_{ep}$: 30 | $lr$: 1e-4; $\lambda_{\mathrm{KL}}$: 25; $g_{ep}$: 30 |
| PILoRA | $lr$: 1e-1; $lr_{pr}$ 1e-4 | $lr$: 1e-1; $lr_{pr}$: 1e-4 | $lr$: 1e-1; $lr_{pr}$: 1e-4 |
| **LoRM** | $lr$: 1e-2; $r$: 8 $\gamma$: (0, 0.5) | $lr$: 1e-2; $r$: 8 $\gamma$: (0, 0.5) | $lr$: 1e-2; $r$: 4 $\gamma$: (0, 0.5) |

## D.7 CUB-200

| Distrib. $\beta$ | 1.0 | 0.5 | 0.2 |
|---|---|---|---|
| EwC | $lr$: 1e-5 | $lr$: 1e-5 | $lr$: 1e-5 |
| LwF | $lr$: 1e-5 | $lr$: 1e-5 | $lr$: 3e-5 |
| FisherAVG | $lr$: 1e-5 | $lr$: 1e-5 | $lr$: 1e-5 |
| RegMean | $lr$: 1e-5; $\gamma$: (0.1, 0.1) | $lr$: 1e-5; $\gamma$: (0.1, 0.1) | $lr$: 1e-5; $\gamma$: (0.1, 0.1) |
| CCVR | $lr$: 1e-5 | $lr$: 1e-5 | $lr$: 1e-5 |
| L2P | $lr$: 3e-1 | $lr$: 3e-1 | $lr$: 3e-1 |
| CODA-P | $lr$: 1e-3 | $lr$: 1e-3 | $lr$: 1e-3 |
| FedProto | $lr$: 1e-5 | $lr$: 1e-5 | $lr$: 1e-5 |
| TARGET | $lr$: 1e-4; $\lambda_{\mathrm{KL}}$: 25; $g_{ep}$: 30 | $lr$: 1e-4; $\lambda_{\mathrm{KL}}$: 25; $g_{ep}$: 30 | $lr$: 1e-4; $\lambda_{\mathrm{KL}}$: 25; $g_{ep}$: 30 |
| PILoRA | $lr$: 1; $lr_{pr}$ 1e-4 | $lr$: 1; $lr_{pr}$: 1e-4 | $lr$: 1; $lr_{pr}$: 1e-4 |
| **LoRM** | $lr$: 1e-2; $r$: 1 $\gamma$: (0, 0.3) | $lr$: 3e-2; $r$: 1 $\gamma$: (0, 0.3) | $lr$: 3e-2; $r$: 1 $\gamma$: (0, 0.3) |

## D.8 OUT-OF-SCOPE

| Distrib. $\beta$ | 1.0 | 0.5 | 0.2 |
|---|---|---|---|
| RegMean | *lr*: 3e-3; $\gamma$: (0.5, 0.5) | *lr*: 3e-3; $\gamma$: (0.5, 0.5) | *lr*: 3e-3; $\gamma$: (0.5, 0.5) |
| CCVR | *lr*: 3e-4 | *lr*: 3e-4 | *lr*: 1e-3 |
| Ties-Merging | *lr*: 1e-3 | *lr*: 1e-3 | *lr*: 1e-3 |
| **LoRM** | *lr*: 1e-2; *r*: 2 $\gamma$: (0, 0.5) | *lr*: 1e-2; *r*: 1 $\gamma$: (0, 0.5) | *lr*: 1e-2; *r*: 1 $\gamma$: (0, 0.1) |

