# OpenReview forum: "Closed-Form Merging of Parameter-Efficient Modules for Federated Continual Learning"
_ICLR.cc/2025/Conference — ICLR 2025 Poster_

### Official Review · Reviewer_AkSa · 2024-10-29

**Soundness:** 3
**Presentation:** 3
**Contribution:** 3
**Rating:** 6
**Confidence:** 5

**Summary:**

This paper proposes a solution for federated continual  learning, where information is learned across multiple devices, with each device continually learning sets of disjoint classes. The proposed method proposes a two stage approach. The first stage involves learning LoRA modules in an alternating cycle, where one of the A or B matrices is kept frozen at a time while the other is optimized across individual devices. Once this has been done for all tasks across all devices, the LoRA matrices are aggregated one final time using RegMean to create one model with knowledge from each device across all tasks. The proposed method outperforms presented CIL and FCIL

**Strengths:**

- Adequate novelty is shown by introducing a training scheme that can properly update LoRA modules across multiple clients.

- LoRM preserves privacy between devices by using Gram matrices

- Ablations clearly show the importance of the proposed components.

- Provided results show significant improvement over SOTA.

- Extremely detailed list of parameters for all experiments is provided, greatly aiding replication.

- Very straightforward organization, the paper flows logically and is easy to read.

**Weaknesses:**

- The paper mentions  that there is relatively little work in FCIL, a brief diagram visualizing the problem setup would greatly aid in understanding the problem.

- Figure diagrams, namely Figure 1 requires more detailed captions. - The current SOTA for exemplar free CIL, RANPAC, is missing from the comparison list, this method also uses Gram matrices collecting information across tasks, so a comparison where it is adapted to be a FCIL method would be relevant.

- The Previous SOTA, PILoRA shows results for both Quantity-based label imbalance and Distribution-based label imbalance, but results in this paper are only shown for Distribution-based.

**Questions:**

- Replicated results appear to be conflicting with previous works (L2P and TARGET’s relative ranking is switched in the PILoRA paper). Is there a reason for this discrepancy?

- It is unusual to see L2P outperforming Coda-P, as Coda-P outperforms L2P in the standard CIL setting. What is the reason for this? How do RANPAC (https://arxiv.org/abs/2307.02251) and DualPrompt (https://arxiv.org/abs/2204.04799) compare? Does this inverse relationship between CIL accuracy and FCIL accuracy hold?

- ImageNet-A, CUB, and Cars196 have been shown to be more challenging datasets for CIL methods, with wider gaps in accuracy when comparing techniques. How does LoRM perform on these datasets?

---

> ### Author Response · Authors · 2024-11-22
>
> Below, we summarize reviewer AkSa's concerns and respond to each one comprehensively
>
> **1. A diagram visualizing the FCIL setting would be helpful**
> To better clarify the Federated Class-Incremental Learning (FCIL) setting, we will include a diagram that visually represents this framework. Given the space constraints, we will reference this diagram in the main paper and provide the full illustration in the appendix.
>
> **2. More detailed caption for figure 1**
> We have updated the caption of Figure 1 to improve clarity. We hope that the revised version is more understandable. We report it here for convenience.
>
> > Training and aggregation procedure of LoRM. For a generic layer, at the end of the communication round $j$ or $j+1$, we obtain the global $B_M$ or $A_M$ matrix, respectively, starting from *i)* the distributed $A$'s or $B$'s, and *ii)* the gram matrices ($XX^{\top}$'s).  $B_M$ or $A_M$ will serve as the fixed matrix in the next round. Finally, we apply RegMean incrementally to compute $\Delta W$.
>
> **3. Missing competitors: RanPAC and DualPrompt**
> We conducted additional experiments involving these two methodologies on CIFAR-100 and Imagenet-R, and provide the results below.
>
> ### CIFAR-100
> |$\beta$|0.5|0.1|0.05|
> |-|-|-|-|
> |RanPAC|**85.89**|77.65|62.84|
> |DualPrompt|53.32|21.89|19.56|
> |LoRM|83.19|**82.05**|**81.21**|
>
> ### Imagenet-R
> |$\beta$|0.5|0.1|0.05|
> |-|-|-|-|
> |RanPAC|63.60|51.30|47.02|
> |DualPrompt|41.17|24.80|13.91|
> |LoRM|**75.72**|**70.83**|**70.18**|
>
> **4. Quantity-based label imbalance**
> Additional experiments on the quantity-based label imbalance are provided in the following for the CIFAR-100 dataset. The dataset choice is motivated by CIFAR-100 being a common benchmark between our work and PILoRA. In the table, $\alpha$ indicates the number of classes in the quantity-based scenario. Following PILoRA, we used 6, 4, and 2.
>
> ### CIFAR-100
> |$\alpha$|6|4|2|
> |-|-|-|-|
> |EWC|62.45|62.03|49.65|
> |LwF|47.08|45.17|30.13|
> |DER++|61.80|59.94|49.11|
> |L2P|58.96|45.33|32.83|
> |CODA-P|29.97|16.68|8.90|
> |FisherAvg|60.49|62.69|50.85|
> |CCVR|79.82|82.10|78.71|
> |PILoRA|76.68|74.78|71.29|
> |LoRM|**82.61**|**82.29**|**82.20**|
>
>
> **5. L2P and TARGET’s relative ranking is switched w.r.t. PILoRA's experiments**
> The reviewer correctly observed that, while L2P outperforms TARGET in our experiments, the opposite result is reported in the PILoRA paper. Although this may seem counterintuitive, we would like to highlight key differences between our experimental setup and that of PILoRA. Specifically, our setting uses fewer communication rounds (5 vs. 30) and a different pretraining checkpoint (ImageNet-21k vs. DINO). In PILoRA's scenario, the extended 30-rounds training may increase forgetting in L2P, as it lacks an explicit regularization mechanism to mitigate it. In contrast, TARGET's distillation approach likely helps preserving knowledge over the longer sequence of rounds.
>
> **6. L2P surprisingly outperforms CODA-Prompt**
> In the original Class-Incremental Learning setting, where both methodologies were introduced, CODA-Prompt generally outperforms L2P. However, in our decentralized experimental setting, the dynamics can differ. We have no theoretical intuition for such behavior but we have thoroughly checked the source code of both methodologies, and successfully replicated the results reported in their respective original papers within their specified settings.
>
> **7. Additional results for ImageNet-A, CUB, and Cars196**
> We conducted additional experiments on these datasets, and report the results below.
>
> ### CUB
> |$\beta$|1.0|0.5|0.2|
> |-|-|-|-|
> |EWC|46.03|39.25|41.97|
> |LwF|42.61|42.72|13.17|
> |DER++|54.63|49.34|48.55|
> |L2P|64.24|60.63|51.16|
> |CODA-P|37.56|30.19|22.63|
> |FisherAvg|43.17|45.18|39.51|
> |CCVR|56.02|52.05|55.37|
> |TARGET|18.26|15.45|13.17|
> |PILoRA|60.96|60.44|61.08|
> |LoRM|**66.78**|**64.43**|**61.10**|
>
> ### CARS
> |$\beta$|1.0|0.5|0.2|
> |-|-|-|-|
> |EWC|29.21|30.15|27.75|
> |LwF|22.60|29.21|24.42|
> |DER++|36.82|36.30|32.83|
> |L2P|32.72|21.24|25.38|
> |CODA-P|2.91|2.67|1.79|
> |FisherAvg|22.65|20.64|25.07|
> |CCVR|16.76|16.39|15.81|
> |TARGET|12.30|9.12|0.63|
> |PILoRA|37.40|37.35|**37.06**|
> |LoRM|**37.61**|**41.46**|36.22|
>
> ### Imagenet-A
> |$\beta$|1.0|0.5|0.2|
> |-|-|-|-|
> |EWC|12.44|11.45|13.03|
> |LwF|12.64|9.69|10.6|
> |DER++|17.58|14.94|6.65|
> |L2P|24.03|19.55|16.00|
> |CODA-P|15.80|13.56|7.44|
> |FisherAvg|16.09|16.92|10.73|
> |CCVR|28.70|25.28|29.3|
> |TARGET|3.42|3.03|2.76|
> |PILoRA|21.53|21.99|21.53|
> |LoRM|**41.61**|**39.3**|**35.02**|

---

> > ### Comment · Reviewer_AkSa · 2024-11-25
> > **Response to rebuttal.**
> >
> > Thanks for you rebuttal, I am satisfied with your response to all of my questions. I am happy to increase the score.

---

> > > ### Author Response · Authors · 2024-11-26
> > >
> > > We sincerely thank the reviewer for their thoughtful feedback on our work and for acknowledging our responses to all their questions.
> > >
> > > We are especially grateful for the reviewer’s willingness to increase the score from the initial 6. However, it seems that the score has not been updated yet. We kindly wanted to check if this might have been an oversight and, if so, would deeply appreciate their consideration in revising it.
> > >
> > > Thank you again for your valuable time and insights.

---

> > > ### Author Response · Authors · 2024-11-30
> > > **Gentle reminder**
> > >
> > > Dear Reviewer AkSa, if your opinion remains unchanged from your last message, we kindly ask you to officially raise your score from the initial 6, as previously committed.
> > >
> > > Such an adjustment would greatly contribute to the final decision and would mean a lot to us. Thank you for your understanding and support.

---

> > > > ### Author Response · Authors · 2024-12-04
> > > > **Gentle reminder**
> > > >
> > > > Dear Reviewer AkSa, are you still committed to **raising your score** from the initial 6? If so, we ask you to make it official, as it would **greatly influence** the final decision. Thank you!

---

### Official Review · Reviewer_BHoB · 2024-11-04

**Soundness:** 2
**Presentation:** 2
**Contribution:** 1
**Rating:** 5
**Confidence:** 5

**Summary:**

Low-rank Regression Mean (LoRM) is a novel method to merge LoRA (Low-Rank Adaptation) modules in a closed-form solution specifically designed for Federated Class-Incremental Learning (FCIL) by alternatively freezing and merging A and B submatrices of LoRA method.

**Strengths:**

The method is privacy preserving and enables continual merging for the Image Classification experiments shown.

**Weaknesses:**

**Inadequate Experimental Diversity** LoRM builds on LoRA, a parameter efficient finetuning method which was introduced to finetune large LLMs and other large, computationally expensive models for diverse and complex tasks. However, LoRM is only tested on the task of image classification and on a single backbone. This counteracts the motivation of using LoRAs (and LoRM is specific to only LoRAs). Experiments with different backbones - LLM, text-image models and diverse tasks like commonsense reasoning, text to image generation - where LoRA modules are regualrly - are necessary to show the efficacy of LoRM, especially since the contributions in the work are backed by only empirical results. LoRM's effectiveness across other federated learning tasks also remains unclear. Testing LoRM in other tasks, such as reinforcement learning or generative tasks, would strengthen its generalizability​.

**Sensitivity to Data Distribution**: LoRM’s performance may degrade with extreme data heterogeneity. The method could benefit from further optimization or regularization to ensure robustness across highly imbalanced data distributions typical in federated settings​.

It is unclear why the alternating freezing is needed? Why not train another LoRA which combines the previous base model+ LoRA weights? That appears to be the simplest solution to the model merging problem and would avoid problems like highly imbalanced data distributions across different tasks which could lead to load imbalance on particular matrices. Ablation for this would also have been helpful.

Comparisons to model merging and lora merging methods like [1, 2] are needed since this method focuses on this very aspect.

[1] Stoica et al. - ZipIt! Merging Models from Different Tasks without Training
[2] Yadav et. al - TIES-Merging: Resolving Interference When Merging Models

**Questions:**

Please refer to the questions in the weakness section

---

> ### Author Response · Authors · 2024-11-22
>
> We thank reviewer BHoB for their valuable insights. Below, we provide detailed responses to their comments and concerns.
>
> **1. Inadequate experimental diversity**
> Regarding the experimental scope, we chose image classification as a foundational task to establish the efficacy of LoRM in a well-studied domain. We acknowledge the importance of evaluating LoRM on more diverse tasks and backbones, such as large language models (LLMs). We are actively working on additional experiments on intent classification (Out Of Scope, OOS [3]); they will be provided by the end of the rebuttal period whether possible. Otherwise, we will add them to the final version of the manuscript. Additional settings such as text-to-image models, other federated learning settings, reinforcement learning and generative tasks represent exciting avenues which we will explore in future works.
>
> The reviewer also suggested that our contributions are backed by only empirical results. While our current work primarily focuses on image classification, we would like to emphasize that our contributions are not solely empirical. Specifically, we provide theoretical results that extend merging to different parameter-efficient fine-tuning (PEFT) techniques, including **LoRA** (appendix A1), **VeRA** (appendix A2), and **IA3** (appendix A3), demonstrating the broader applicability of our method. Additionally, we prove that concatenating the incremental heads is mathematically equivalent to minimizing the RegMean regression problem across all incremental tasks (appendix A4).
>
> **2. Sensitivity to data distribution**
> We thank the reviewer for suggesting additional regularization objectives, which could potentially improve performance in scenarios characterized by high data heterogeneity.
>
> Even without incorporating extra regularizations -- which could increase the computational and memory footprint of LoRM -- we achieve state-of-the-art results under the highest levels of data heterogeneity reported in the literature (i.e., dirichlet distribution $\beta = 0.05$). As also requested by reviewer AkSa, we have included additional results for quantity-based label imbalance to further address this aspect and provide a clearer understanding of LoRM's performance in such conditions. The reviewer can find these results in the response to reviewer AkSa.
>
> **3. It is unclear why the alternating freezing is needed**
> If we understand correctly, the reviewer suggests that reinitializing a new LoRA module at each round, starting from the full-rank merged module of the previous round, would achieve comparable performance while being simpler.
>
> Although this suggestion is valid, implementing such a technique would severely increase the communication cost by a factor of $18.9\times$, as it requires sharing the full-rank updates back to the clients at each communication round. This becomes a critical limitation in Federated Learning, as would compromise the algorithm's feasibility in decentralized scenarios.
>
> **4. Comparisons to model merging [1, 2]**
> We thank the reviewer for the suggestion, acknowledging the importance of comparing our method to existing model merging approaches such as ZipIt! [1] and TIES-Merging [2]. However, ZipIt! is not directly applicable to the Federated Learning scenario, as it requires access to model activations, which are not available for the server model (that is responsible for merging the clients' updates).
>
> Instead, TIES-Merging operates solely on the parameters, allowing its applicability to our scenario. We will update the experimental section accordingly, including this method by the end of the rebuttal period.
>
> [1] Stoica, G., et al. (2023). Zipit! merging models from different tasks without training. In *ICLR*.
>
> [2] Yadav, P., et al (2024). Ties-merging: Resolving interference when merging models. In *NeurIPS*.
>
> [3] Larson, S., et al. (2019). An evaluation dataset for intent classification and out-of-scope prediction. In *EMNLP-IJCNLP*.

---

> ### Comment · Reviewer_BHoB · 2024-11-24
> **Response**
>
> 1. **Inadequate experimental diversity** Thank you for the response. I'd like to point the paper claims to be a general method for parameter efficient training for "federated learning" which it underlines by showing applicability to LoRA. As we know LoRA is and has been applied to a large number of domain and modalities. Therefore, showing results only on Image Classification and on 3 conventional datasets doesn't satisfy the initial claim. It'd been a different case if the paper has claimed a narrow application to only image classification, but in that case the use of LoRA, which was initially created as a method efficiently fientune large LLMs would have been problematic.
>
> 2. **Sensitivity to data distribution** Although I appreciate authors' EuroSat results, as pointed out in the previous paragraph, as per the claims of the current version of the paper, I find a single experiment regarding OOD robustness not fully adequate.
>
> 3. **It is unclear why the alternating freezing is needed** I don't think I agree with the authors here, however if communication compression or efficiency is an issue, why not simply transfer the modification of the current LoRA matrix from a client in the form of a rank update or another low rank factorization of $\Delta A$ or $\Delta B$. Such *hierarchical* LoRA variants actually exist in literature and more efficient than simply transferring individual matrices.
>
> 4. **Comparisons** As noted in above paragraph, if the main objective is communication efficiency as the authors claim, then comparisons with model merging and more efficient PeFT methods is needed.

---

> > ### Author Response · Authors · 2024-11-25
> >
> > **1. Inadequate experimental diversity**
> >
> > We remark that the paper is about Federated Class-Incremental Learning, and should be compared with existing literature of the topic. Indeed, all competitors perform experiments analogous to us, or even on less datasets. Moreover, in reply to reviewer AkSa, we added **three** more datasets (i.e., CARS, CUB, and Imagenet-A), making our submission the one with the vastest experimental campaign on FCIL. We will be extremely happy if the reviewer can point out other FCIL works that comply to their idea of *satisfying experimental diversity*. We advocate that, although we may agree with the reviewer about testing the solution on many different scenarios, their concern is more related to the literature on the topic than to our specific submission. Moreover, we are committing the best of our efforts to include an experiment on document classification.
> >
> > **2. Sensitivity to data distribution**
> >
> > As per the previous response, our experimental protocol is in line with the literature of the topic -- i.e. Federated Class Incremental Learning, and **not** Federated Learning in general. Moreover, as remarked before, we added **three** more datasets that exhibit significant diversity w.r.t. the pretraining.
> >
> > **3. It is unclear why the alternating freezing is needed**
> >
> > We are sorry for not being completely clear in the first response. We gave for granted the main motivation of the alternated optimization, which is strictly related to the merging problem we aim to solve. As outlined in section 3.1, merging the LoRA matrices cannot be achieved in a closed form due to the resulting underdetermined system of equations. To address this, we adopted alternated optimization as a solution -- a widely used technique in various optimization contexts, including adversarial learning. By doing so, we can help minimizing the communication cost, which is a plus for FCIL but not our main objective. We appreciate the reviewer's suggestions (hierarchical LoRA) but without extensive empirical validation we cannot say for sure that it is a feasible and working solution. We will be very happy if the reviewer can point out a FCIL paper that leverages such a solution. On the contrary, our proposal has been empirically validated and we can say with a good certainty that it is effective in the referenced scenario.
> >
> > **4. Comparisons with model merging and more efficient PeFT methods is needed**
> >
> > Our experimental analysis includes PEFT techniques, specifically L2P and CODA-Prompt, which utilize prompting, and PILoRA, which employs LoRA. Additionally, in response to reviewer AkSa's suggestion, we incorporated DualPrompt, another prompting methodology. This brings the total to **four** competitors utilizing PEFT techniques.
> >
> > As a general remark, we do agree that probably our solution does not perfectly fit the taste of the reviewer. Nevertheless, we are politely asking the reviewer if they really find it of such a low quality to be scored as a clear rejection.

---

> ### Comment · Reviewer_BHoB · 2024-11-27
> **Official Comment by Reviewer**
>
> 1. I'd like to note that my contention is with the overtly broad claims regarding federated learning made in your work. Your work is Federated Class-Incremental Learning for Image Classification only. However, your paper appears to present it as a general FCIL work, leaving the scope of the work highly ambiguous. This ambiguity is amplified by the usage of LoRA, which was initially introduced to efficiently finetune large language models. Therefore, I find that the currently written paper is too broad in its claims and need to be rewritten to adequately define the correct scope. Even after that, the question arises as to why use LoRA for a task and a backbone which is not afflicted by the problem of efficiency in the first place[1], and why claim a closed form solution for LoRA in general while only showing a "non-large" model backbone for experiments, and on datasets which do not require extensive compute or time to train. In summary, the paper makes too broad claims in its current form and does not explain well why LoRA should be used in this highly narrow, constrained manner.
>
> 2. I'd stand by my earlier comment - as per the claims of the current version of the paper, I find a single experiment regarding OOD robustness not fully adequate, and even the current one does not provide a good evidence for claims of robustness that the authors write.
>
> 3. There are number of methods which would provide more "communication efficiency" over transferring entire LoRA matrices, such as [2,3,4,5,6,7]. A few of these papers are unpublished as of now, and there is no need to compare to them - however, they underline the general idea of a whole area of very efficient low rank adapters which freeze or share different parts of the low rank matrices which can be trivially applied to the FCIL tasks especially since the "communication efficiency" here is important. These papers do not even include Mixture of Expert models, where one would only need to transfer sparse expert weights among the nodes of a federated system. The problem is that this paper introduces a solution for a problem without evaluating whether currently available methods can do similarly well or even better - there is no theoretical reasoning or empirical results as to why the closed form solution would be better than the other relevant PeFT or mixture methods for better efficiency and performance, especially since the focus is on *efficiency*.
>
> 4. I'd also argue that the current paper, with its only focus on image classification (relatively small datasets at that) and a single, relatively lightweight backbone does not provide any practical benefits to the community in terms of federated continual learning. Federated learning, among other motivations, is used to make learning of large models, more energy and compute efficient by using shared resources. In case of a ViT, I'd argue that more energy would be spent sending and receiving data than training the lightweight model on a single modern GPU. As such, the current experimental setup remains largely theoretical and does not hold much practical value for the community. This would change if the paper showed results on larger models, and compute extensive and complex tasks, like Natural Language Generation, Image Generation, etc.
>
> As a general remark, I believe the current version of the paper makes unproven broad claim with regards to federated continual learning as a whole. It also uses low rank adaptation, which originally developed for finetuning large models efficiently, for a single task and backbone which arguably do not share the same efficiency problem for what LoRA was originally developed for. Further, using LoRA, worsens the problem of the unproven claims as the paper gives off an idea that this method works beyond IID image classification task when there is no evidence otherwise - the current version of the paper does not clearly disambiguate this in their paper or the title. Further since the main contribution is "communication efficiency", there exist a wide host of current methods which are good solutions to this problem setup, and the authors do not provide an effective reason or results as to why their method is better especially since the focus is on data, compute and communication efficiency. Further the current experimental setup holds little practical value for the community (point 4). Until these points are remedied, I believe my rating is a fair evaluation.
>
> [1] Nag et al. - ViTA: A Vision Transformer Inference Accelerator for Edge Applications
>
> [2] VeRa: Vector-based Random Matrix Adaptation
>
> [3] Balazy et al. - LoRA-XS: Low-Rank Adaptation with Extremely Small Number of Parameters
>
> [4] SVFT: Parameter-Efficient Fine-Tuning with Singular Vectors
>
> [5] MiLoRA: Harnessing Minor Singular Components for Parameter-Efficient LLM Finetuning
>
> [6] Batched Low-Rank Adaptation of Foundation Models
>
> [7] LaMDA: Large Model Fine-Tuning via Spectrally Decomposed Low-Dimensional Adaptation

---

> > ### Author Response · Authors · 2024-11-29
> >
> > > Overly broad claims, experiments on just image classification and relatively lightweight backbone
> >
> > The existing FCIL literature is **exclusively** focused on image classification, leading us to adopt this **standard** protocol outlined in prior works [8, 9, 10]. Similarly, we include evaluation against continual learning methods adapted to FCIL, all of which also restrict their experiments to **image classification** only.
> >
> > Furthermore, we hereby present the results of our experiments on **text classification** using the sequential Out-of-Scope dataset [11], demonstrating that our methodology achieves SOTA performance even with larger textual backbones such as T5 [12]. The following table shows the results across 10 tasks, with 15 classes per task, and 10 clients.
> >
> > |Distribution $\beta$|1.0|0.5|0.1|
> > |-|-|-|-|
> > |RegMean|44.09|42.67|37.29|
> > |Ties Merging|57.96|54.84|53.76|
> > |CCVR|66.47|70.11|71.51|
> > |LoRM|**81.82**|**77.36**|**72.33**|
> >
> > > Why use LoRA for a task and a backbone which is not afflicted by the problem of efficiency
> >
> > Many Federated Learning [10, 13, 14] and Continual Learning [15, 16, 17] works leverage LoRA on ViTs, remarking that this improves compositional capabilities, improving performance, while also boosting efficiency.
> >
> > > The paper does not explain well why LoRA should be used
> >
> > We leverage LoRA for its compositional capabilities and communication efficiency properties, as highlighted in the abstract and elaborated in Sections 3 and 4. We will revise the manuscript to clarify this aspect.
> >
> > > I find a single experiment regarding OOD robustness not fully adequate
> >
> > We did provide the results for **three additional** OOD datasets in response to rewiewer AkSa. This makes up the total OOD datasets to **four** (i.e., Eurosat, CUB-200, Cars-196, Imagenet-A).
> >
> > > Here are number of methods which would provide more "communication efficiency" over transferring entire LoRA matrices, such as [2,3,4,5,6,7]
> >
> > We respecfully remark that **none** of these methods is tailored to either FL, FCIL or Model Merging, and **four out of six** are both not published and contemporary to our work. For one of the two that are published [2], we already provided theoretical results (closed-form merging solution) in the appendix of our manuscript.
> >
> > > This paper introduces a solution for a problem without evaluating currently available methods
> >
> > Our experimental analysis comprises **10** competitors, **4** of which are based on PEFT. In this rebuttal, we have included **three** additional methods: DualPrompt, RanPAC, and Ties-Merging. The effectiveness and extensiveness of our experimental setting is also acknowledge by other reviewers.
> >
> > > No theoretical reasoning or empirical results as to why the closed form solution would be better for efficiency and performance
> >
> > Our primary focus is on achieving optimal performance while maintaining efficiency. The proposed layer-wise merging technique represents the **optimal solution** to the problem defined in Section 2, which was formulated with LoRA matrices to explicitly account for efficiency. Empirically, we provide a thorough experimental comparison, as aknowledged by other reviewers, that shows the effectiveness of our proposal.
> >
> > > The paper gives off an idea that this method works beyond IID image classification
> >
> > **None** of the evaluated scenarios follow an IID distribution. Instead, all experiments are conducted across varying levels of distribution imbalance (i.e., $\beta \in \{1.0, 0.5, 0.2, 0.05\}$, including the most extreme cases reported in the literature ($\beta = 0.05$).
> >
> > Finally, we thank the reviewer for giving us the opportunity to improve our work. The discussion turned out to be **constructive**, leading us to introduce new experiments and insights that **enhance the paper**.
> >
> > [2-7] As per the previous comment from the reviewer
> >
> > [8] Yoon, J., et al. (2021). Federated continual learning with weighted inter-client transfer. In *ICML*.
> >
> > [9] Zhang, J., et al. (2023). Target: Federated class-continual learning via exemplar-free distillation. In *ICCV*.
> >
> > [10] Haiyang, G., et al. (2024). Pilora: Prototype guided incremental lora for federated class-incremental learning. In *ECCV*.
> >
> > [11] Larson, S., et al. (2019). An evaluation dataset for intent classification and out-of-scope prediction. In *EMNLP-IJCNLP*.
> >
> > [12] Raffel, C., et al. (2020). Exploring the limits of transfer learning with a unified text-to-text transformer. In *JMLR*.
> >
> > [13] Kuo, K., et al. (2024). Federated LoRA with Sparse Communication. ArXiv preprint.
> >
> > [14] Bian, J., et al. (2024). LoRA-FAIR: Federated LoRA Fine-Tuning with Aggregation and Initialization Refinement. ArXiv preprint.
> >
> > [15] Gao, Q., et al. (2023). A unified continual learning framework with general parameter-efficient tuning. In *ICCV*.
> >
> > [16] Wistuba, M., et al. (2023). Continual learning with low rank adaptation. In *NeurIPS workshop*.
> >
> > [17] Chitale, R., et al. (2023). Task Arithmetic with LoRA for Continual Learning. In *NeurIPS workshop*.

---

### Official Review · Reviewer_H9U5 · 2024-11-06

**Soundness:** 3
**Presentation:** 3
**Contribution:** 3
**Rating:** 8
**Confidence:** 4

**Summary:**

This paper studies the model merging, specifically focusing on merging the low-rank adaptation (LoRA) modules. The author proposed to extend the work of RegMean to merge LoRA modules in a closed form, where the challenge lies in the infeasibility of solving the indeterminate system introduced by the LoRA modules. The author further proposes an alternative optimization algorithm, i.e., freezing one of the two matrices, A and B, in the LoRA modules and optimizing each alternatively. The author chose the federal class incremental learning as the case study for the proposed method, conducted extensive experiments to verify the effectiveness of proposed method.

**Strengths:**

1. The extension of RegMean for merging the LoRA module is interesting. Given the extensive applicability of the LoRA for parameter-efficient fine-tuning for foundation models, the reviewer believes that the proposed method may benefit a broad range of applications in the future.

2. The proposed alternative optimization procedure for extending RegMean to Low-rank Regression Mean (LoRM) is intuitive and sound. It is promising that the algorithm can converge quickly in practice, as shown in Fig. 2, compared to the methods that achieved the state-of-the-art (SOTA) results previously.

3. The chosen testbed, i.e., the federal class incremental learning (FCIL), is a surging research direction in continual learning with real-world board applications. The author conducted extensive comparisons with different methods and showed the superiority of the LoRM, which vastly improved the SOTA of FCIL.

4. The code has been attached for replication, which will benefit future study.

**Weaknesses:**

1. Since the proposed method is general for model merging, the reviewer may want to see whether it can achieve good results in different applications, e.g., parameter-efficient fine-tuning and class-incremental learning in computer vision and/or natural language processing.

2. Although the author shows that the LoRM can converge quickly in practice, the reviewer still wonders whether there are any theoretical guarantees for its convergence.

**Questions:**

Please refer to the Weaknesses section.

**Details Of Ethics Concerns:**

N/A.

---

> ### Author Response · Authors · 2024-11-22
>
> Below, we address reviewer H9U5's questions and concerns in detail.
>
> **1. Additional results on class-incremental Learning (Computer Vision and Natural Language processing)**
> Our experimental analysis focuses on incremental computer vision tasks (classification) learned in a decentralized fashion. Regarding computer vision, we have conducted additional experiments on other datasets as requested by reviwer AkSa (i.e. CUB-200, CARS, Imagenet-A), and we report the results in the response to AkSa. We acknowledge that further analysis on natural language processing tasks could be valuable. We are actively working on this and will include the results by the end of the rebuttal period if possible; otherwise, they will be incorporated into the final version of the manuscript.
>
> **2. Theoretical guarantees for convergence**
> As this point was also raised by reviewer rSBD, we have provided the derivation of the convergence proof for LoRM in response to their rebuttal. Due to space constraints, we refer the reviewer to that response without duplicating the proof here.
>
> While LoRM has been experimentally shown to converge faster than other competitors, we do not yet have a theoretical proof to explain this behavior. We hypothesize that this improvement arises from two key factors: i) LoRM utilizes low-rank fine-tuning via LoRA, which reduces both global and local gradient magnitudes; ii) our layer-wise merging technique facilitates a better solution to the global optimization problem w.r.t. simple averaging, further decreasing the aforementioned gradients.

---

> > ### Comment · Reviewer_H9U5 · 2024-11-25
> > **Thank you for the response**
> >
> > My concerns have been resolved, and I am happy to raise my score.

---

> ### Author Response · Authors · 2024-11-26
>
> We sincerely thank the reviewer for raising their score and for their thoughtful feedback on our work. We appreciate their recognition of our responses to their concerns and remain available to address any further questions.

---

### Official Review · Reviewer_rSBD · 2024-11-07

**Soundness:** 3
**Presentation:** 3
**Contribution:** 3
**Rating:** 6
**Confidence:** 3

**Summary:**

The paper presents a novel optimization method for Federated Class-incremental Learning (FCIL). Based on the LoRA approaches, the authors propose a optimization method to estimate the two low-rank matrices effiently. Rather than the previous approaches upon the various types of assumptions, the proposed algorithm employed the alternating optimization methods by optimizing the two low-rank matrices separately by fixing the other one. The optimization methods was evaluated in the scenario of FCIL, which shows the state-of-the-art performance for CIFAR-100 and ImageNet-R dataset even with the various evalutation settings.

**Strengths:**

The target problem and the contribution are well presented in the paper, and the following mathetmatical derivation is also easy to follow. The experimental results verify the effectiveness of the proposed algorithm, and futher ablation studies are well designed to prove the component-wise effectiveness and the mentioned contributions.

**Weaknesses:**

Even with the above mentioned strengths, I have several concerns about the technical algorithm and evaluations as follows:

- Contribution (Alternative optimization vs. closed-form):
 The authors emphasized the contribution of this paper as the alternating optimization and the closed-form optimization. However, the relationship between the two different optimization methods is not precisely described. As I understood, the closed-form optimization is performed at each iteration for the respective low-rank matices, and the overall optimization solver follows the mechanism of alternating optimization. Since the two optimization approaches are definitely different to each other, the authors should specify the relationship of the two contributions.

- Convergence check for the optimization solver:
 When we design the optimization solver, it is important to check its convergence. When the optimization fails to convergence, the solution can be diverged, which causes serious problems. If the convergence check fails, the strategy to stop the iteration should be given. In the paper, both the convergence check and the stop criterion are not described.

- Validity of alternating optimization:
 The approach of alternating optimization is one of the most common optimization solvers. When we employ the alternating optimization approach by fixing one other of two parameters, it is important to check the relationship between the two parameters. When the two paraemters are not indenpendent to each other, the iteration with separated optimization can cause serious problem where the basic assumption for the two parameter can be broken. To prevent the issue, the conventional alternating optimization employs a penalty term where the relationship between two optimizing parameters can be preserved as the iteration goes. I hope to know that the proposed algorithm considers the relationship between the two low-rank matrices or the two low-rank matrices are independent to each other.

- Additional measurements:
 The quantitative results are present only by the average accuracy for entire classes. However, to evaluate the FCIL method, tthe measurements of he average forgetting and accuracy of all seen classes in the final task are meaningful. To show the effectiveness of the proposed algorithm for the last rounds and the final round, I recommend such additional measurements in the tables.

- Comparison fairness:
 It seems that the most recent study (PILoRA) shows low performance upon ImageNet-R dataset. Of course, the performance can be dramatically dropped in the new dataset, but it shows too large gap between its reported performance with the ImageNet-based dataset. I hope to know the reason why the large gap happens.

**Questions:**

The validity of the optimization solver should be checked.
The lack of evaluation metrics should be supplemented to show the effectiveness of the proposed algorithm.

---

> ### Author Response · Authors · 2024-11-22
> **Part 1**
>
> Below, we summarize reviewer rSBD's concerns and provide detailed responses.
>
> **1. Contribution (Alternative optimization vs. closed-form)**
> We agree with the reviewer that alternating the training of the A and B LoRA matrices is a standalone technique that can be applied independently; however, in our approach, this alternation is inherently tied to the closed-form merging technique we employ for FCIL. In fact, the merging problem defined in Equation 5 cannot be solved in a single closed-form manner for both A and B simultaneously, as the system is underdetermined (Equation 6). To address this, we alternate the optimization of these two variables, treating one as constant while solving for the other, thereby finding a determinate solution. As correctly understood by the reviewer, closed-form merging is applied at each communication round (i.e. the iteration of the global algorithm) for the respective low-rank matices, and the overall optimization solver follows the mechanism of alternating optimization.
>
> **2. Convergence check for the optimization solver**
> For what concern the convergence check, we refer the reviewer to [1], which demonstrates the convergence for a federated algorithm that applies the FedAVG technique. Differently from FedAVG, our approach changes the update rule, leveraging the optimal solution to the regression problem for each layer (equation 5 of the manuscript) instead of simple parameter averaging. Formally, for a generic **even** round $j$:
>
> \begin{equation}
> \overline{\Delta}^{j} = B_M^{j} A_M^j = B_M^{j} \left(\sum_{i = 1}^m A_i^j X_i X_i^\top \right) \left(\sum_{i = 1}^m X_i X_i^\top\right)^{-1},
> \end{equation}
>
> where $m$ denotes the number of clients, $\overline{\Delta}^j$ the full-rank update for the round $j$, $B$ and $A$ represent the LoRA matrices, with the subscript $_M$ standing for *merged*, and $X_i X_i^\top$ corresponds to the Gram matrices of the inputs.
> As we are alternating the optimization, for a generic **odd** round $j$:
>
> \begin{equation}
> \overline{\Delta}^{j} = B_M^j A_M^{j} = \left(\sum_{i = 1}^m B_i^j A_M^{j} X_i X_i^\top\right) A_M^{j\top} \left(A^{j} \sum_{i = 1}^m X_i X_i^\top A_M^{j\top} \right)^{-1} A_M^{j}.
> \end{equation}
>
> Following the convergence proof of [1] (appendix A.1, equation 2, second-last line), with $W$ indicating the full-rank parameters for clarity, we can rewrite the central term $\frac{\eta_L}{2m}\sum_{i=1}^{m}\sum_{k=0}^{K-1}\mathbb{E}||\nabla F_i(W_{j, k}^i) - \nabla F_i(W_j)||^2$ as:
>
> \begin{equation}
> \frac{\eta_L}{2m}\sum_{i=1}^{m}\sum_{k=0}^{K-1}\mathbb{E}\left|\left|B_M^{j}\left[\nabla F_i(A^i_{j, k}) - \nabla F_i(A_j)\right] X_i X_i^\top \left(\sum_{l = 1}^m X_l X_l^\top\right)^{-1}\right|\right|^2, \text{ and}
> \end{equation}
>
> \begin{equation}
> \frac{\eta_L}{2m}\sum_{i=1}^{m}\sum_{k=0}^{K-1}\mathbb{E}\left|\left|\left[\nabla F_i(B^i_{j, k}) - \nabla F_i(B_j)\right]A_M^{j}X_i X_i^\top A_M^{j} \left(\sum_{l = 1}^m A_M^{j} X_l X_l^\top A_M^{j} \right)^{-1} A_M^{j}\right|\right|^2,
> \end{equation}
>
> for the **even** and **odd** rounds respectively. The same modification applies to the first term of *appendix A.1 of [1], equation 3, last line*. Similarly, we replace the second term of the same line $\frac{\eta^2_L}{m^2}\mathbb{E}\left|\left|\sum_{i=1}^{m}\sum_{k=0}^{K-1}\nabla F_i(W^i_{j, k})\right|\right|^2$ as:
>
> \begin{equation}
> \frac{\eta^2_L}{m^2}\mathbb{E}\left|\left|\sum_{i=1}^{m}\sum_{k=0}^{K-1}B_M^{j-1}\nabla F_i(A^i_{j, k})X_i X_i^\top \left(\sum_{l = 1}^m X_l X_l^\top\right)^{-1}\right|\right|^2, \text{ and}
> \end{equation}
>
> \begin{equation}
> \frac{\eta^2_L}{m^2}\mathbb{E}\left|\left|\sum_{i=1}^{m}\sum_{k=0}^{K-1}\nabla F_i(B^i_{j, k})A_M^{j}X_i X_i^\top A_M^{j} \left(\sum_{l = 1}^m A_M^{j} X_l X_l^\top A_M^{j} \right)^{-1} A_M^{j}\right|\right|^2.
> \end{equation}
>
> In each round, our solution introduces a dependency on the input data of each client by leveraging the corresponding Gram matrix as a weighting factor. This factor is normalized by multiplying it with the inverse of the sum of all Gram matrices across all clients. Since the Gram matrix is positive semi-definite, this normalization ensures that the resulting norm remains bounded. Building on Assumption 3 from [1] (bounded local gradients, $\sigma_L$, and global gradients, $\sigma_G$) and leveraging Lemma 4 from [2], these expectations can be reformulated in terms of the constants $\sigma_L$ and $\sigma_G$. We will include the complete proof in a new section of the appendix.
>
> [1] Yang, H., et al. (2021). Achieving linear speedup with partial worker participation in non-iid federated learning. In *ICLR*.
>
> [2] Reddi, S., et al. (2021). Adaptive federated optimization. In *ICLR*.

---

> > ### Author Response · Authors · 2024-11-22
> > **Part 2**
> >
> > **3. Validity of alternating optimization**
> > Alternating optimization is a widely used approach for solving optimization problems. However, in our work, this technique is employed solely to address the initial regression problem (equation 5 of the manuscript), rather than to enhance convergence properties of our algorithm. The two variables are specifically introduced to enforce low-rank updates, without altering the convergence rate of the overall optimization process.
> >
> > In detail, the gradient with respect to the full-rank matrix $W$ is projected using a fixed $A$ or $B$, depending on the communication round. The update $\Delta B$ is expressed as $\sum_{i=0}^{K} \nabla F(W_k) A^{\top}$, while the update $\Delta A$ is given by $\sum_{i=0}^{K} B^{\top} \nabla F(W_k)$. This process inherently makes the two matrices interdependent, as each serves to project the full-rank gradient $\nabla F(W_k)$ through the other.
> >
> > However, since the gradient is independent of the two matrices, the rows of $A$ and the columns of $B$ remain linearly independent. As a result, the concatenation of $A$ and $B$ would still be full-rank. We hope this explanation clarifies our approach and the theoretical intuition underlying the alternate optimization. If further clarification is needed, we would be glad to provide additional comments.
> >
> > **4. Additional measurements:**
> > We thank the reviewer for suggesting forgetting as an additional measurement. Given that we have already initiated numerous experiments to address other reviewers' concerns, we will re-run the experiments and include the results incorporating forgetting by the end of the rebuttal period. These will be included in the Appendix of the final version of the manuscript.
> >
> > Regarding the accuracy across all classes, we kindly ask the reviewer for clarification. Specifically, we are uncertain whether the comment refers to multiple accuracy values corresponding to the number of seen classes (which would be 100 numbers for CIFAR-100, or 200 for ImageNetR) or a single accuracy value representing the average accuracy across all classes. If it is the latter, the results we reported already reflect this approach; otherwise, we would be happy to provide the detailed accuracy values for all classes.
> >
> > **5. Comparison fairness**
> > It is true that PILoRA demonstrates higher performance on TinyImageNet and ImageNet-200 (a subset of ImageNet comprising 200 classes) in the original paper compared to ImageNet-R in our work. Although this may seem unusual, there is a significant distinction between TinyImageNet, ImageNet-200, and ImageNet-R. Specifically, the first two datasets are subsets of ImageNet, whereas ImageNet-R has a different distribution, as it is a classification problem focused on sketches. We argue that this is the main reason for the performance gap that the reviewer rightly noticed. Moreover, the authors of PILoRA use a different experimental setting, which involves a higher number of communication rounds (30 instead of 5).

---

> > > ### Author Response · Authors · 2024-11-27
> > >
> > > Dear reviewer rSBD, we deeply value your feedback and wanted to ensure that you’ve had the opportunity to review our responses to your comments.
> > >
> > > If there are any additional concerns or clarifications needed, we would be more than happy to address them promptly. Your input is highly appreciated, and we look forward to your response.
> > >
> > > Thank you once again for your time and consideration.

---

> ### Comment · Reviewer_rSBD · 2024-11-28
>
> The meanings of convergence check and some of the previously confused terms have been clarified after the rebuttal. I hope the authors will revise the paper to better explain the definitions provided in the comments. Specifically, I expect the convergence check and additional measurements to be included in the appendix or elsewhere. Believing that these points will be addressed, I have decided to increase my score.

---

> > ### Author Response · Authors · 2024-11-29
> >
> > We appreciate the reviewer’s thoughtful feedback and are glad that our rebuttal helped clarify the key points of our work. If there are any further questions or concerns, we would be happy to address them.
> >
> > In the revised manuscript, we will report the convergence check and the additional metrics on all datasets, to ensure a more comprehensive analysis.

---

### Author Response · Authors · 2024-11-22

We sincerely thank all the reviewers for their valuable feedback and constructive comments. We are particularly pleased that reviewers **rSBD, H9U5, and AkSa** appreciate the logical flow and soundness of our methodology and manuscript, generally describing it as well-presented. We further express our gratitude to reviewers **rSBD and H9U5** for acknowledging the experimental section as effective, with reviewer **H9U5** highlighting its extensive nature, and to reviewers **rSBD and AkSa** for recognizing the efficacy of the ablation study.

We are also delighted that reviewers **BHoB and AkSa** recognize the privacy-preserving capabilities of **LoRM**, with reviewer **AkSa** additionally acknowledging the novelty of our approach. Furthermore, we appreciate that reviewers **rSBD, H9U5, and AkSa** underline the achievement of state-of-the-art results in Federated Class-Incremental Learning, with reviewer **H9U5** emphasizing the importance of our proposed testbed. Reviewer **H9U5** also noted the broader impact of the manuscript, which we deeply value.

---

> ### Public Comment · ~Jiao_Chen3 · 2025-02-19
>
> I truly appreciate your insightful work on this topic. There are also other works that follow a somewhat similar recursive approach, which might complement and enrich the literature review. Although not focused on federated continual learning, works such as [1] and [2] adopt a closed-form solution with pre-trained CLIP, which could provide additional context to your discussion. It might be helpful to include these in the camera-ready version to complete enrich the literature review.
>
> Thank you again for your thoughtful work on this topic.
>
> [1] ACIL: Analytic class-incremental learning with absolute memorization and privacy protection. NeurIPS 2022.
>
> [2] Advancing Cross-domain Discriminability in Continual Learning of Vison-Language Models. NeurIPS 2024

---

### Meta-Review · Area_Chair_nPCp · 2024-12-17

**Metareview:**

The paper introduces LoRM, a technique for model merging Federated Continual Learning that builds upon low-rank adaptation methods like LoRA. The authors propose to unify models by ensuring that the merged model matches the responses of all participating models. The authors address the challenge of indeterminate solutions through an alternating optimization strategy, which allows for training one LoRA matrix at a time. The method is applied to the Federated Class-Incremental Learning (FCIL) scenario and it demonstrates state-of-the-art performance in a variety of FCIL scenarios.

Reviewers variously praised the submission for its clear presentation of the target problem and contributions, as well as its easy-to-follow  mathematical derivations and overall logical organization of the paper. The majority of reviewers found the experimental results -- and especially the ablations -- to convincingly support the effectiveness of LoRM. The extension of RegMean to merge the LoRA module is considered by most reviewers to be interesting and potentially beneficial for a variety of applications. The alternating optimization algorithm for LoRM shows quick convergence compared to the previous state-of-the-art, and the paper addresses a relevant research (Federated Class Incremental Learning) and demonstrates significant improvement over existing methods.

Reviewer rSBD raised a number of critical issues in their review, particularly regarding the clarity of the relationship between alternating and closed-form optimization, a lack of discussion of convergence checks and the overall validity of the alternating optimization approach, and questions about fairness of comparative evaluations. The author rebuttal provided additional details regarding convergence of the alternating optimization method and clarifications regarding performance on ImageNet-R which presents a very large domain shift compared to other benchmark datasets. The authors should include these details regarding convergence in the final version of the paper.

Reviewer BHoB criticizes the paper for its lack of experimental diversity, since it concentrates exclusively on image classification tasks. In rebuttal, the authors correctly observe that the vast majority of work on FCIL concentrates on image recognition and thus the paper aligns with the current state-of-the-art (and thus ensures broad comparability with existing methods). The authors also provide additional experimental results on text classification which demonstrate the effectiveness of LoRM on text backbones. It is the AC's opinion that the positive and novel aspects of the contribution outweigh this limitation and the decision is to Accept.

**Additional Comments On Reviewer Discussion:**

There was significant back-and-forth between several reviewers and the authors during the discussion phase. Reviewer rSBD raised a number of critical issues in their review, particularly regarding the clarity of the relationship between alternating and closed-form optimization, a lack of discussion of convergence checks and the overall validity of the alternating optimization approach, and questions about fairness of comparative evaluations. The author rebuttal provided additional details regarding convergence of the alternating optimization method and clarifications regarding performance on ImageNet-R which presents a very large domain shift compared to other benchmark datasets.

Reviewer BHoB criticizes the paper for its lack of experimental diversity, since it concentrates exclusively on image classification tasks. In rebuttal, the authors correctly observe that the vast majority of work on FCIL concentrates on image recognition and thus the paper aligns with the current state-of-the-art (and thus ensures broad comparability with existing methods). The authors also provide additional experimental results on text classification which demonstrate the effectiveness of LoRM on text backbones.

It is the AC's position that the novel contributions and solid experimental validation outweigh the limitations articulated by reviewers (in particular Reviewer BHoB). The claims in the paper are not overstated, and concentrating on a specific CIL domain is common if not practically necessary.

---

### Decision · Program_Chairs · 2025-01-22

Accept (Poster)